# The intensification of Arctic warming as a result of CO$_2$ physiological forcing

So-Won Park [1], Jin-Soo Kim [2,3✉] & Jong-Seong Kug[1✉]

Stomatal closure is one of the main physiological responses to increasing CO$_2$ concentration, which leads to a reduction in plant water loss. This response has the potential to trigger changes in the climate system by regulating surface energy budgets—a phenomenon known as CO$_2$ physiological forcing. However, its remote impacts on the Arctic climate system are unclear. Here we show that vegetation at high latitudes enhances the Arctic amplification via remote and time-delayed physiological forcing processes. Surface warming occurs at mid-to-high latitudes due to the physiological acclimation-induced reduction in evaporative cooling and resultant increase in sensible heat flux. This excessive surface heat energy is transported to the Arctic ocean and contributes to the sea ice loss, thereby enhancing Arctic warming. The surface warming in the Arctic is further amplified by local feedbacks, and consequently the contribution of physiological effects to Arctic warming represents about 10% of radiative forcing effects.

[1] Division of Environmental Science and Engineering, Pohang University of Science and Technology (POSTECH), Pohang, South Korea. [2] School of GeoSciences, University of Edinburgh, Edinburgh, UK. [3] Department of Evolutionary Biology and Environmental Studies, University of Zurich, Zurich, Switzerland. ✉email: jinsoo.kim@uzh.ch; jskug@postech.ac.kr

The increase in atmospheric $CO_2$ concentration has an influence on plant physiology. Physiological responses to increasing $CO_2$ include changes in leaf area index (LAI) and stomatal conductance, and those affect the plant transpiration in opposite ways. First, one of the main physiological responses is the $CO_2$ fertilization effect—that is, an increase in the rate of photosynthesis[1–3]. This effect accounts for the largest contribution to the positive trends in the LAI detected by satellite data sets[4] and can also lead to an increase in plant transpiration, resulting in cooling effects[5]. Second, another plant response is a reduction in stomatal conductance. In other words, stomatal apertures open less widely under elevated $CO_2$ concentrations. These $CO_2$-induced reductions in the stomatal conductance have been confirmed through experiments and from reconstruction data[2,3,6,7].

The stomatal closure resulting from elevated $CO_2$ levels can decrease the rate of transpiration by diminishing the amount of water loss from plants. This reduction in plant transpiration can lead to an increase of near-surface air temperature by decreasing the evaporative cooling effect and simultaneously increasing the sensible heat flux above the land surface[8–10]. This nonradiative effect from physiological acclimation is known as $CO_2$ physiological forcing[11]. Previous studies using model experiments have investigated how the physiological forcing will affect the future climate in vegetation-covered regions, through influences such as amplified heat extremes[12], intensified zonal asymmetry of rainfall over tropical land[13], drying over the Eastern Amazon[14] and Sahel greening[15].

This physiological effect has a potential to remotely alter the entire climate system through the redistribution of the surface energy and disturbance of hydrological cycle, but still, the remote impacts of physiological effect on the climate system are unclear especially in the Arctic region (north of 70°N). The Arctic is the region most sensitive to greenhouse warming and has experienced warming faster than the global average, a phenomenon known as Arctic amplification[16]. Many mechanisms have been suggested to explain the Arctic amplification including a role of diminishing sea ice[17,18], seasonal storage and release of the absorbed shortwave (SW) radiation coupling with sea-ice loss[19–21], enhanced downward longwave (LW) radiation due to an increase in water vapor and cloud fraction[22,23], ocean biogeochemical feedback[24,25], increased poleward energy transport[26,27] and other processes[28,29]. However, their relative contributions are still under debate and also many alternative mechanisms are under investigation. Here, we suggest that the $CO_2$ physiological forcing has a remote forcing on the Arctic climate and can intensify the Arctic amplification through the enhanced atmospheric poleward heat transport and the physical processes coupling with the Arctic sea-ice change.

To examine the impacts of physiological acclimation under elevated $CO_2$ on the future climate system, we analyzed eight Earth system models (ESMs), which can simulate interactions between the physical climate system and the biogeochemical processes, from the Coupled Model Intercomparison Project Phase 5 (CMIP5)[30] (Supplementary Tables 1 and 2). In line with previous studies[12,13,31–34], we respectively quantified the physiological forcing (Phy), which includes the $CO_2$ fertilization effect and the dependency of stomatal conductance on $CO_2$, and $CO_2$ radiative forcing (Rad) (average $CO_2$ concentrations ~823 ppm) using carbon–climate feedback experiments (see the Methods section and Supplementary Table 3).

## Results
**Land surface warming by plant physiological effects.** Figure 1 shows changes in the annual mean evapotranspiration (ET), Bowen ratio (the ratio of sensible to latent heat fluxes), and near-

**a**

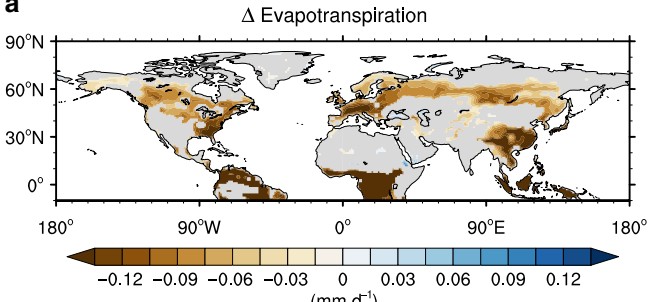

Δ Evapotranspiration

**b**

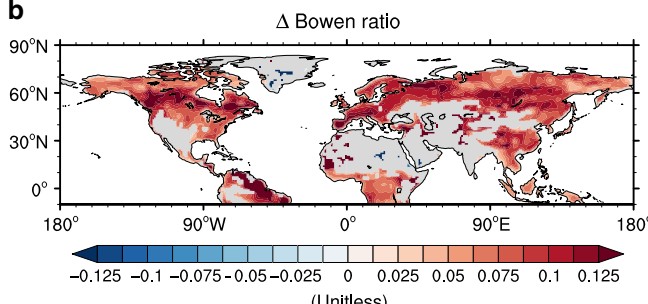

Δ Bowen ratio

**c**

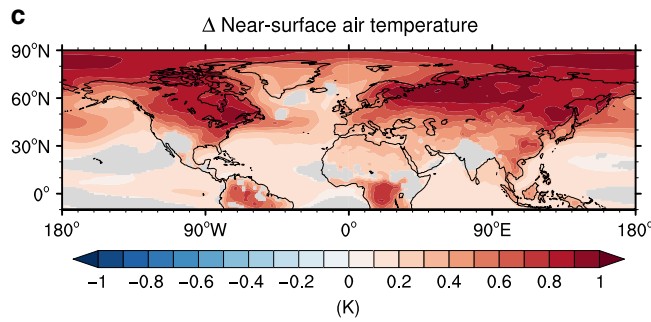

Δ Near-surface air temperature

**Fig. 1 Change in evapotranspiration, Bowen ratio and temperature resulting from $CO_2$ physiological forcing.** Multimodel mean change of the annual mean evapotranspiration (**a**), Bowen ratio (sensible heat flux/latent heat flux) (**b**), and near-surface air temperature (**c**) resulting from $CO_2$ physiological forcing. Only significant values at the 95% confidence level based on a bootstrap method are shown.

surface air temperature resulting from the $CO_2$ physiological forcing. In contrast with the radiative effect inducing the increase in ET due to enhanced water-demand from the temperature rise (Supplementary Fig. 1), physiological effects cause a conspicuous and significant reduction in the annual mean ET in densely vegetated areas of the tropics and mid-to-high latitudes (Fig. 1a) in line with previous studies[12,31,32,34–37]. In this idealized experiment for evaluating the $CO_2$ physiological forcing, the fertilization effect plays a role in increasing ET due to the resulting increased LAI, but the effect of stomatal closure works in the opposite way at the same time[5,10,37]. Therefore, this overall drop in ET suggests that the stomatal closure have a greater influence in controlling the total ET than the $CO_2$ fertilization, when only the physiological effects is considered under elevated $CO_2$ levels, in consistency with the argument in previous studies[12,31,34,37].

The physiological effects change the surface energy budgets by reducing the evaporative cooling and simultaneously increasing the sensible heat flux (Fig. 1b and Supplementary Table 4). These heat flux changes induce surface and near-surface air warming around regions where ET is significantly decreased. Interestingly, significant surface warming occurs in the Arctic Ocean under the influence of $CO_2$ physiological forcing, despite the fact that

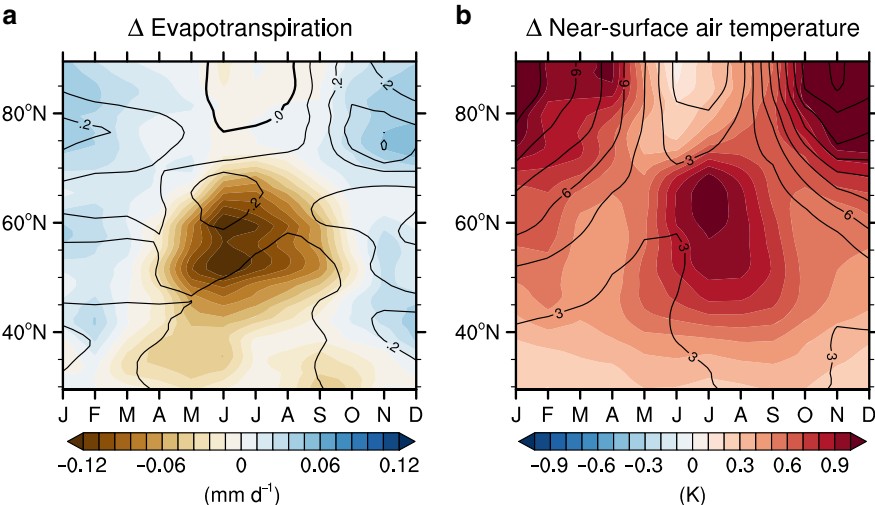

**Fig. 2 Impacts of the physiological forcing on the evapotranspiration and temperature.** Zonally and monthly averaged change in the evapotranspiration (**a**) and surface air temperature (**b**). The shading represents the change resulting from $CO_2$ physiological forcing. The contouring represents the change resulting from $CO_2$ radiative forcing. The contour intervals for radiative forcing are 0.1 mm day$^{-1}$ in (**a**) and 1.5 K in (**b**).

vegetation obviously does not exist in the Arctic Ocean (Fig. 1c). Another interesting point is that a synergy effect, a nonlinear interaction of physiological forcing with the radiative forcing[15,37], additionally contributes to the surface warming (see the Methods section and Supplementary Table 5). The magnitude of synergy effect in Arctic region is equivalent to ~24% of annual mean temperature change resulting from physiological forcing. These results imply that the global warming signal by the radiation forcing plays a role in amplifying the physiological effect through their interactions. Meanwhile, the physiological forcing excluding a synergy effect still induces the statistically significant Arctic warming, which confirms the consistency and robustness of our findings (Supplementary Fig. 2).

Seasonal changes caused by the physiological effects (Fig. 2 and Supplementary Fig. 3) were further examined. While the variations of ET remain almost constant throughout the year in tropical regions, the reduction in ET at mid-to-high latitudes (40°–70°N) exhibits a strong seasonality. As a result, the changes in the surface energy fluxes and temperature in mid-to-high latitudes also show a strong seasonality. In summer (June–July–August; JJA) when photosynthesis is most active, the maximum decline in ET and the resulting strongest continental warming occur. Unlike this continental warming (40°–70°N), however, the maximum warming in the Arctic regions due to physiological effects, an increase of +0.99 K, occurs in winter with a time lag (Supplementary Table 6). The mechanisms of this remotely induced Arctic warming are discussed in the next section.

**Arctic warming remotely induced by $CO_2$ physiological forcing**. As illustrated in Fig. 1, changes in plant physiology lead to a statistically significant temperature rise in the Arctic Ocean. The continental warming in JJA resulting from the physiological responses seems to be propagated to the polar region with time (Fig. 2b, shading), whereas this pattern was not observed in the $CO_2$ radiative forcing experiment (Fig. 2b, contour). The Arctic warming resulting from the physiological effects is most distinctive during the boreal winter (December–January–February; DJF). In addition, the magnitude of this Arctic warming in DJF is comparable with that of continental warming during JJA.

It is important to understand how this continental warming resulting from the physiological effects can remotely cause the distinctive delayed warming in the Arctic Ocean. A previous study

demonstrated that mid- and high-latitude forcing can remotely contribute to the Arctic warming through various physical processes[38]. In particular, the increase in atmospheric northward energy transport (NHT$_{ATM}$) due to the continental warming can be responsible for the delayed Arctic warming. It is evident that the northward energy transport significantly increases during the warm season from April to July (see the Methods section and Supplementary Fig. 4a). Furthermore, this NHT$_{ATM}$ continuously increases during the whole period of simulations due to the intensified $CO_2$ physiological forcing with increasing $CO_2$ levels (Supplementary Fig. 4b). These results imply that the NHT$_{ATM}$ plays a role in connecting the extratropical continental warming to the Arctic warming under an influence of the physiological effect, which shows the similarity with previous studies, suggesting that the high-latitude greening and mid-latitude afforestation can enhance the Arctic amplification through an increase in poleward energy transport[39–42]. The increase in NHT$_{ATM}$ is associated with sea-ice melting and the resultant newly open waters in the Arctic allow it to absorb more sunlight during the warm season (Supplementary Table 6). Most of this energy is released to the atmosphere through the longwave radiative flux, and sensible and latent heat flux in the Arctic Ocean during autumn and winter, thereby inducing the Arctic warming (Supplementary Table 6). These mechanisms, the seasonal storage and release of the absorbed shortwave radiation coupling with an Arctic sea-ice change, have already been proposed in previous studies to explain the Arctic amplification[19–21]. Nonetheless, our results suggest that the plant physiological forcing as well as radiative forcing can contribute to the Arctic amplification under elevated $CO_2$ levels.

A previous study has shown that the mid-troposphere in the Arctic sensitively responds to the energy advection across the Arctic boundary[26]. The bottom-heavy warming profile has been attributed to increased upward turbulent heat fluxes by the loss of sea ice in previous studies[17,19]. The vertical structure of atmospheric warming shows that the mid-tropospheric warming first occurred with a large vertical extent in the Arctic region during summer, and then this warming was propagated to the lowermost region of the atmosphere with time (Supplementary Fig. 5). These results support our hypothesis that the remote and lagged effects of plant physiological acclimation can intensify Arctic warming through an enhancement of NHT$_{ATM}$ and the resulting Arctic sea-ice change. However, there is a large inter-model diversity in the magnitude of Arctic warming, which seems

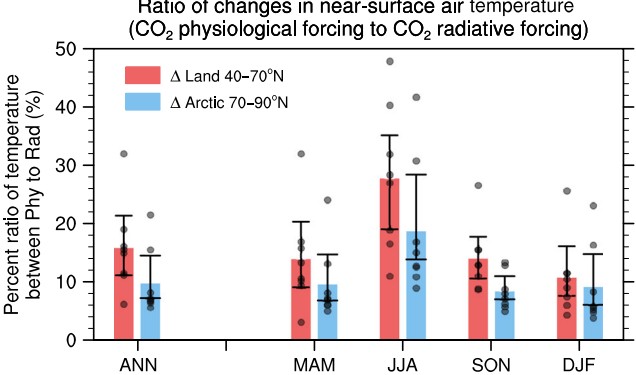

Ratio of changes in near-surface air temperature
($CO_2$ physiological forcing to $CO_2$ radiative forcing)

**Fig. 3 Ratio between changes of temperature in response to $CO_2$ physiological forcing and radiative forcing.** Ratio of the change in the near-surface air temperature resulting from $CO_2$ physiological effects to that resulting from $CO_2$ radiative forcing ([$CO_2$ physiological forcing/$CO_2$ radiative forcing] × 100) in the continents (40°–70°N) and Arctic region (70°–90°N), respectively. Each bar shows the area-weighted average of multimodel ensemble. The black dots represent the individual results from ESMs. The error bar for each column indicates the range of the 95% confidence level on the basis of a bootstrap method.

to be closely related to the strength of local feedback related to Arctic sea ice, but further research is needed to confirm this (Supplementary Figs. 6 and 7).

In summary, the surface warming resulting from physiological effect enhances an atmospheric energy convergence into the Arctic basin and this increases the net SW absorption during the warm season in the Arctic Ocean by melting sea ice. Subsequent energy release to the atmosphere increases the air temperature and ice-free waters in the Arctic, thereby intensifying the Arctic amplification during the cold season. As a result, the $CO_2$ physiological forcing accounts for 27.7% of the continental warming in summer and 9.7% of the annual surface warming in Arctic region resulting from $CO_2$ radiative forcing (Fig. 3). These results emphasize that the contribution of the plant physiological effects to the Arctic warming is quite significant.

**Intensified and continued surface warming by local feedback.** Besides the direct heating from the enhanced sensible heat flux, an increase in net shortwave absorption (4.58 W m$^{-2}$ in JJA) additionally heats the air above the land surface in JJA (Supplementary Table 4). In this experimental design, the net SW absorption can be largely affected by these two factors: An increase in LAI resulting from $CO_2$ fertilization effect can alter the surface albedo and increase the net SW absorption, thereby contributing to the temperature rise. The decrease in cloud fractions caused by physiological acclimation-driven reduction of relative humidity[35,43,44] can also be a cause of surface warming because it enhances downward SW radiative flux[42]. From their relative contributions, we found that vegetation-cloud feedback has a dominant role in the increased net SW absorption during summer (Supplementary Fig. 8), thereby contributing the continental warming (40°–70°N) (Fig. 4 and Supplementary Fig. 9) particularly in summer (Supplementary Table 4). Furthermore, the relative magnitude of the vegetation-cloud feedback in ESMs seems to explain the inter-model diversity of the land surface warming (40°–70°N) in JJA ($r = -0.79$, $P = 0.02$) (Supplementary Fig. 7). Specifically, two models, HadGEM2-ES and MPI-ESM-LR, show the greatest warming in JJA due to this greatest cloud effect despite the moderate reduction of ET (Supplementary Figs. 10–12).

In contrast to the change of cloud cover over the continents (40°–70°N), the cloud formation is enhanced in the Arctic region

especially during winter (Fig. 4 and Supplementary Table 6). This increased cloud fraction additionally intensifies the surface warming by decreasing the outgoing longwave radiation especially in non-summer season[45,46] (Supplementary Table 6). Although it is difficult to prove the causality in this experiment, it is conceived that this increase in cloud formation contributes to the Arctic sea-ice loss, which in turn causes the increase in water vapor from the newly opened Arctic waters, as proposed previously[21,46]. In summary, the cloud feedback in the Arctic can enhance the surface warming by increasing a downward LW radiation, and in turn, the enhanced surface warming can accelerate the sea-ice loss, thereby causing positive feedback during the cold season.

Another local feedback might be triggered by physiological forcing over the continents (40°–70°N). As shown in Fig. 4, a snow concentration and a surface albedo in high latitudes significantly decline in response to the $CO_2$ physiological forcing. The warming resulting from the physiological effects presumably melts snow and the resultant less-snow-covered surface absorbs more solar radiation (Supplementary Table 4). Furthermore, an increase of LAI from the fertilization effects and the land cover change in models with interactive vegetation might partially contribute to the surface warming by altering the surface albedo independently of a change in temperature and would melt the snow as noticed previously[47,48] (Supplementary Figs. 13 and 14). Consequently, this snow–albedo feedback may help enhance and maintain the land surface warming throughout the year, especially in high latitudes where the surface albedo is relatively high due to the high snow cover (Fig. 4). On the whole, our results suggest that the local feedbacks triggered by physiological effects might additionally contribute to the amplified and maintained surface warming in both continents and Arctic Ocean.

**Discussion**
So far, it has been shown based on a multimodel mean that distinctive Arctic warming occurs due to the physiological effects, but this conclusion can be model-dependent because the structure of models and the parameterization schemes are different from each other. The magnitudes and spatial patterns of change in ET and temperature are diverse and HadGEM2-ES seems to greatly contribute to the multimodel ensemble mean temperature change (Supplementary Figs. 10 and 11). Nevertheless, most models consistently simulate the reduction in ET, the resulting surface warming over the continents and enhanced Arctic warming as a result of physiological effect (Supplementary Fig. 6), which suggests that the results are not sensitive to a subsampling of the models. In addition, multimodel ensemble results excluding HadGEM2-ES are not much different with those including HadGEM2-ES and still statistically significant though the magnitude is a bit altered (Supplementary Fig. 15). These again attest to the robustness of our findings and also suggest that ensemble mean is not controlled by an outlier.

This study shows that the physiological effects amplify Arctic warming by 9.7% compared with that from the radiative forcing. This surface warming in the Arctic region resulting from physiological response might have the potential ramifications of future changes in the carbon and hydrological cycles by intensifying the interaction between the Arctic climate and Arctic biological system[47,48]. Considering the physiological effects of $CO_2$ might be helpful for understanding the inter-model diversity in future climate change. A previous study has reported that the stomatal conductance schemes in the current ESMs do not consider various plant water use strategy[49], which can lead to the underestimation of the surface warming across Northern Eurasia[50]. This result raises a possibility that Arctic warming may be greater than that in the current projections. Furthermore, there are still the

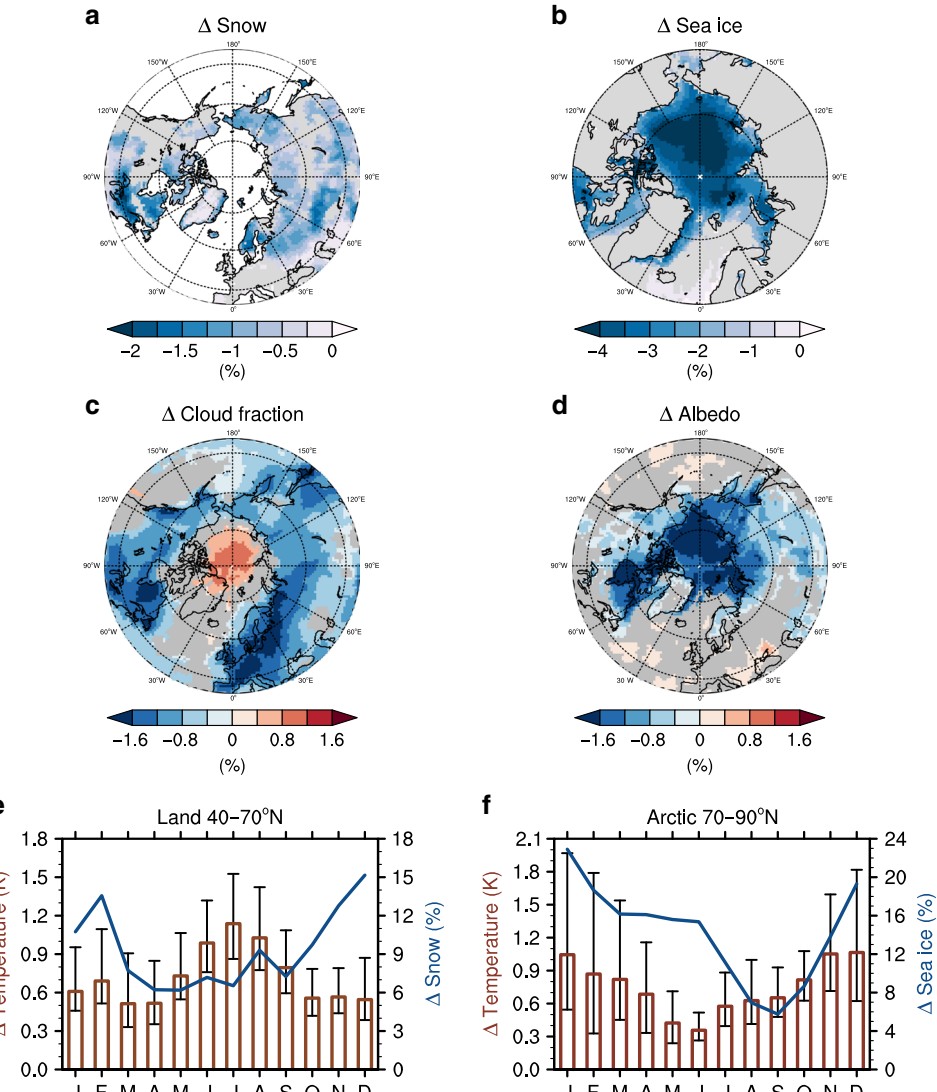

**Fig. 4 Local feedback triggered by CO$_2$ physiological forcing.** Multimodel mean change of the annual mean snow concentration (**a**), sea-ice concentration (**b**), total cloud fraction (**c**) and surface albedo (**d**). Only significant values at the 95% confidence level based on a bootstrap method are shown. **e, f** Annual cycle of change in the surface air temperature (red bar) resulting from CO$_2$ physiological effect with the ratio between change in the snow concentration (blue line) in response to physiological forcing and radiative forcing in continents (40°–70°N) (**e**). The same as in (**e**), but in the Arctic region (70°–90°N), with the ratio between change in the sea-ice concentration (blue line) in response to physiological forcing and radiative forcing (**f**). The error bars represent a range of the 95% confidence level on the basis of a bootstrap method. All of the values are area-weighted averages of eight ESMs, except for the snow concentration, which is an average of five ESMs, because GFDL-ESM2M, HadGEM2-ES, and IPSL-CM5A-LR do not provide the surface snow area fraction data.

limitations of land surface models in simulating LAI and the albedo dynamics and the stomatal conductance schemes in ESMs are rather static and semi-empirical (see the Supplementary Notes 1 and 2). These factors make it hard to simulate the realistic plant behavior to elevated CO$_2$ levels and also increase the uncertainty in the quantification of climate change caused by the physiological forcing. These point to the need for improvement of land models' schemes based on a fundamental understanding of the involved processes.

## Methods

**CMIP5 data and experimental design.** Eight ESMs (bcc-csm1-1, CanESM2, CESM1-BGC, GFDL-ESM2M, HadGEM2-ES, IPSL-CM5A-LR, MPI-ESM- LR, and NorESM1-ME) were used, which were coupled with the full carbon cycle and used in idealized experiments designed to assess carbon–climate feedback, from the CMIP5 archive[30] (see Supplementary Tables 1 and 2). These experiments were run for 140 years with 1% per year increase in atmospheric CO$_2$ concentration from

preindustrial levels to quadrupling (285−1140 ppm) for both radiation and bio-geochemistry (1pctCO2), radiation only (esmFdbk1) and biogeochemistry only (esmFixClim1) (see Supplementary Table 3). In GFDL-ESM2M, the atmospheric CO$_2$ levels were prescribed to increase from their initial mixing ratio level of 286.15 ppmv at a rate of 1% per year until year 70 (the point of doubling, 2 × CO$_2$) and thereafter CO$_2$ concentrations were kept at a constant for the remainder of the run.

To quantify the CO$_2$ physiological forcing (Phy) (average CO$_2$ concentrations ~823 ppm), we calculated the difference between the final 70 years of two simulations data: Full CO$_2$ simulation (1pctCO2) that includes the fully interactive radiative, physiological, and fertilization effects in response to increasing CO$_2$ and radiation simulation (esmFdbk1) that includes only radiative effects in response to increasing CO$_2$. Following the previous study[12], we extracted the physiological forcing (Phy) from full CO$_2$ simulation rather than using the physiology simulation directly to evaluate CO$_2$ physiological forcing (Phy) relative to future CO$_2$ radiative forcing (Rad). Since a nonlinear interaction between CO$_2$ radiative forcing and physiological forcing exists in full CO$_2$ simulation, the physiological forcing (Phy), defined as 1pctCO2-esmFdbk1, includes this nonlinear interaction, or synergy effect, as well as the pure physiological forcing. We additionally assessed the physiological forcing in a different way by calculating the difference between the average of the final 70 years of physiology simulation (esmFixClim1) and averaged

values of preindustrial control simulation (piControl) over the whole period to verify the robustness of our finding. Unlike the previous method, this alternative physiological forcing does not include an interaction between physiology and radiation. We evaluated a synergy effect by calculating the difference between the two physiological forcing with the different definition. We quantified the $CO_2$ radiative forcing (Rad) (average $CO_2$ concentrations ~823 ppm) by calculating the difference between the average of the final 70 years of radiation simulation (esmFdbk1) and averaged values of preindustrial control simulation (piControl) over the whole period.

The multimodel ensemble was derived by re-gridding the outputs from ESMs to a common $1° \times 1°$ grid, then averaging together. The bootstrap method was used to test the statistical significance of the difference between the simulations. For MME, eight values were randomly selected from eight ESMs with replacements, and then their average was computed. By repeating this process 1000 times, the confidence intervals were determined, and only significant values were shown to show the model agreement. For each individual model, we randomly selected 70 years with replacements from year 71 to 140, calculated their average and finally computed the confidence intervals by repeating this process 1000 times.

**Atmospheric northward energy transport calculation**. The atmospheric energy convergence into the Arctic basin for transient conditions was estimated using energy budgets and residual methods[51,52]. Following the framework in the previous studies[53,54], the energy budget of an atmospheric column can be denoted as:

$$\frac{\partial E_{ATM}}{\partial t} = NHT_{ATM} + F_{TOA} + F_{SFC}, \tag{1}$$

where $\frac{\partial E_{ATM}}{\partial t}$ is the time change of atmospheric energy storage (W m$^{-2}$), $F_{TOA}$ is the sum of the net radiation budget at the top of atmosphere (W m$^{-2}$), $F_{SFC}$ is the net surface energy flux (W m$^{-2}$) and $NHT_{ATM}$ is the vertically integrated northward heat transport (W m$^{-2}$). All terms are defined as positive when they increase the atmospheric energy, hence positive downward for the TOA net radiation, positive upward for the net surface energy flux and positive for northward heat transport.

Based on Eq. (1), the transient vertically integrated atmospheric northward heat transport can be expressed as:

$$NHT_{ATM} = \frac{\partial E_{ATM}}{\partial t} - F_{TOA} - F_{SFC}. \tag{2}$$

The atmospheric energy storage is written as:

$$E_{ATM} = \frac{1}{g} \int_0^{p_s} \left( c_p T + k + L_v q + \Phi_s \right) dp, \tag{3}$$

where $p$ is pressure (Pa), $p_s$ is the reference surface pressure (hPa), $g$ is gravitational acceleration (m s$^{-2}$), $c_p$ is the specific heat of the atmosphere at constant pressure (J K$^{-1}$ kg$^{-1}$), $T$ is temperature (K), $k$ is the kinetic energy (J kg$^{-1}$), $L_v$ is the latent heat of evaporation (J kg$^{-1}$), $q$ is the specific humidity (kg kg$^{-1}$), and $\Phi_s$ is the surface geopotential which is not a function of pressure[54]. The contribution of kinetic energy, $k$, is ignored here due to its comparatively small magnitude[53].

The net radiation at the TOA, $F_{TOA}$, is defined as:

$$F_{TOA} = F_{SW} - F_{LW}, \tag{4}$$

where $F_{SW}$ is the net shortwave (solar) and $F_{LW}$ is the longwave (thermal) radiation, both in W m$^{-2}$.

The net surface energy budget at the surface, $F_{SFC}$, is defined as:

$$F_{SFC} = SW_{SFC} + LW_{SFC} + Q_H + Q_E, \tag{5}$$

where $SW_{SFC}$ and $LW_{SFC}$ are the net surface shortwave and longwave surface radiative fluxes, and $Q_H$ and $Q_E$ are the net surface sensible and latent heat fluxes, all in W m$^{-2}$.

## Data availability
All CMIP5 data[30] that support the findings of this study are publicly available on Earth System Grid Federation website: https://esgf-node.llnl.gov/.

## Code availability
Processed data, products, and code produced in this study are available from the corresponding authors upon reasonable request.

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

## Acknowledgements

We acknowledge the World Climate Research Programme's Working Group on Coupled Modelling, which is responsible for CMIP, and the climate modelling groups (listed in Supplementary Table 1) for producing and making available their model output. This work was supported by the National Research Foundation of Korea (NRF-2017R1A2B3011511). J.-S. Kim was supported by University of Zurich Research Priority Programme "Global Change and Biodiversity" (URPP GCB).

## Author contributions

S.-W.P. compiled the data, conducted analyses, prepared the figures, and wrote the manuscript. J.-S.Kug and J.-S.Kim designed the research and wrote majority of the manuscript content. All of the authors discussed the study results and reviewed the manuscript.

## Competing interests

The authors declare no competing interests.

## Additional information

**Peer review information** *Nature Communications* thanks Justin Mankin and other, anonymous, reviewers for their contributions to their peer review of this work. Peer review reports are available.

