## [Peer Review File · Nature Communications]

Reviewers' comments:

Reviewer #1 (Remarks to the Author):

Review of The intensification of Arctic warming as a result of CO₂ physiological forcing by Park et al. for Nature Communications:

This study explores the remote impact of CO₂ physiological forcing on Arctic temperatures. The authors use a suite of CMIP5 models that isolate the impact of CO₂ physiological forcing on projected future climate. They find that elevated CO₂ decreases pan Arctic evapotranspiration via reduced stomatal conductance, which increase local temperatures. Via (unspecified) atmospheric transport mechanisms, this heat is carried to the Arctic where it increases temperatures as well. The heat transport then sets off a number of local feedback mechanisms in the Arctic, including reduced sea ice, increased cloud cover, etc. which further promote Arctic warming. The authors also show that CMIP5 models with DGVMs simulate greater Arctic warming in direct response to the CO₂ physiological forcing (warmer temperatures drive vegetation growth and reduced albedo), and speculate that models without DGVMs may underestimate future Arctic warming.

Although vegetation-Arctic-warming mechanisms have been presented in previous literature, the focus on physiological forcing in this study adds a new wrinkle to our understanding, and I think the topic warrants publication in a journal such as Nature Communications. It will be of interested to a broad audience.

However, I suggest the authors address the following comments before the paper should be considered for publication:

Northward Heat Transport Discussion and Supplemental Figure 5:

What are the potential mechanisms for heat transport into the Arctic? Most likely these are atmospheric mechanisms (not ocean).

Additionally, I think there is an error with your energy transport calculation displayed in Supplemental Figure 5. A non-zero transport of energy at the pole is physically impossible. Therefore, a change in the energy transport (as you show) can't be anything but zero.

Additionally, there is a typo in the Figure caption. ϕ in this case is latitude, not longitude.

CO₂ fertilization albedo effects

How does CO₂ fertilization impact the results? You focus on physiological forcing (stomatal conductance), but I expect leaf area index increases from fertilization may also play a role in altering surface albedo and shortwave absorption in all of the models.

Typos/Grammar

There are a number of grammatical mistakes in the paper, including in the Abstract.

Reviewer #2 (Remarks to the Author):

Review for NCOMMS-19-31397 submission from Park et al. entitled "The intensification of Arctic warming as a result of CO₂ physiological forcing."

Park and colleagues leverage the 8 models participating in the C4MIP set of CMIP5-era simulations to show that physiological forcing from CO₂ reduces boreal ET, leading to a partitioning of energy towards sensible heat, thereby amplifying Arctic warming over the land, and in particular, leads to remote warming of surface air over the ocean. The premise of the manuscript is that the remote forcing from stomatal responses is under-explored and the Arctic is a region potentially sensitive to

such forcing.

I have a few comments that I hope the authors will find helpful as they prepare a revision for this or another journal.

1. Generalizability and mechanisms: The premise of the manuscript and its claims to novelty rest on the remote forcing of Arctic temperatures by boreal forest physiological responses to CO₂ and the presentation of a mechanism by which the forcing occurs. The problem is that it is not clear that the remote forcing exists anywhere but in the ensemble mean, nor have the authors actually provided a physical mechanism by which this remote forcing takes place that explains the variability across the models. Much of the basis of the remote forcing claim rests on the results presented in Figs. 2 and S5, which show a time-latitude change in ET and temperature due to Phy and Rad, and the total northward energy transport in summer and annually, respectively. Fig. S6 also contributes to this picture. While it's clear the physiological forcing generates conditions for Arctic warming in the ensemble mean, it's not clear to me that the authors have identified the mechanism, nor is it clear whether such mechanisms exist on a model-by-model basis. Looking at some of the supplementary figures, it appears that a lot of the ensemble mean response over the Arctic is driven by HadGEM2-ES and bcc-csm1-1 in fall and winter (boreal). So it's not clear that the authors are explaining the actual process by which this forcing occurs. For example, Fig S9b shows that the HadGEM2-ES model really drives the remote forcing seen in the ensemble mean plots presented in the main (and it has very little vegetation changes over the regions that purportedly drive the remote forcing, Fig. S13). It would be helpful would be to see the individual energy transport changes for each model (Fig. S5), and to see the changes in cloud fractions at seasonal scales for each model as they present for temperature and ET in the suppmat. Where are the error bars on the bar plot presented in Fig. 4e and f for example? Connecting the ET response to energy partitioning to remote warming (and cloud fractions) across models would actually address this. As of now, I'm not entirely convinced that the remote forcing is a generalized and physically-consistent response within the models due to Phy forcing—it seems to be an outlier story the authors are projecting onto all models.

2. Context relative to Rad-only and Total effects. Some of the work by the authors does present the relative contributions of physiological forcing relative to the Rad-only (e.g., Fig. 2, and 4, etc), the effect size we're talking about here can be quite small, but this reality remains under-emphasized in the manuscript as written. So if there is remote forcing of Arctic, it would be helpful (and interesting) to know if this remote forcing has a nonlinear interaction with the warming from the Rad-only estimate. That is instead of $\text{Phy} + \text{Rad} = \text{Total}$, as assumed in the current form, could the total effects actually be: $\text{Phy} + \text{Rad} + e = \text{Total}$, implying that the total effects can't be linearly decomposed as the authors have done. Something to show that the linearity assumption holds, and that puts the actual Phy effects on Arctic amplification in context fractionally, would be good. It would be really interesting to know if the Phy and Rad are interactive (and nonlinear).

3. The authors claim at line 70 that the ET drop suggests that fertilization effects are smaller than stomatal ones on total ET. This may be true in the Phy-only sense, but the authors have not actually tested it. It is not true when you consider the longer and warmer growing seasons due to Rad interacting with Phy, which leads to a general increase in high-latitude partitioning of precipitation to vegetation and often an increase in ET. So the authors could make sure that is clearer, because they don't actually perform a model-by-model analysis of LAI-driven v. stomatal-driven ET changes. Also, the response as presented in Fig. 1 is really only significant for extremely high latitude vegetation, which is going to be a function of the PFTs the models place there. For example, how much of the ensemble mean ET response is driven by the ET in the IPSL model in summer (Fig. S7b)? If that model were dropped, could the authors still make the claim presented in line 70?

Reviewer #3 (Remarks to the Author):

Summary

This study investigates the role of plant physiological response in the context of rising atmospheric CO₂ for arctic warming amplification. The results are based on a set of CMIP5 climate runs designed to disentangle atmospheric effects (i.e. the radiative global response to elevated CO₂) from the physiological effects (i.e. the vegetation response to elevated CO₂ and its local effect on climate), following an approach very similar to previous studies (e.g. Swann et al. 2016, Skinner et al. 2018, Lemordant et al. 2018, 2019).

In line with previous findings, the authors find that increased plant water use efficiency in boreal regions leads to a net decrease in plant transpiration, thus offsetting an overall increase in LAI (and the associated increase in water demand). This effect is almost exclusively occurring during summer and leads to increased sensible heat, reduced cloud cover, higher net radiation and overall higher surface air temperatures over land. The authors then argue that this additional energy is advected above the arctic ocean, leading to a decrease of sea ice cover and increased cloud fraction, and explaining a delayed winter-time warming over this area revealed by the experiment. The authors calculate that the physiological contribution represents on average 10% of the radiative effect, and thereby positively contributes to the so-called Arctic warming amplification.

Lemordant, L., Gentine, P., Swann, A. S., Cook, B. I., & Scheff, J. (2018). Critical impact of vegetation physiology on the continental hydrologic cycle in response to increasing CO₂. *Proceedings of the National Academy of Sciences*, 115(16), 4093-4098.

Swann, A. L., Hoffman, F. M., Koven, C. D., & Randerson, J. T. (2016). Plant responses to increasing CO₂ reduce estimates of climate impacts on drought severity. *Proceedings of the National Academy of Sciences*, 113(36), 10019-10024.

Skinner, C. B., Poulsen, C. J., & Mankin, J. S. (2018). Amplification of heat extremes by plant CO₂ physiological forcing. *Nature communications*, 9(1), 1094.

Lemordant, L., & Gentine, P. (2019). Vegetation Response to Rising CO₂ Impacts Extreme Temperatures. *Geophysical Research Letters*, 46(3), 1383-1392.

Overall assessment

The CO₂ physiological effects on evaporative demand and their consequences for LAI, surface energy fluxes and air temperature as diagnosed from CMIP5 experiments are relatively well covered by the existing literature. What essentially distinguishes this study from previous work is its focus on the arctic region and its contribution to the scientific discussion on arctic warming amplification. This study is in the line of previous work by some of the coauthors (e.g. Jeong et al. 2014) and demonstrates that physiological effects have a modest (10%) but measurable contribution to arctic warming (independently of changes in vegetation type or cover), thus bringing an incremental but new contribution to our understanding of arctic amplification. The authors make a convincing case, provide good supporting information and the study seems scientifically sound as far as I can judge, with the exception of some points. My main criticism relates to the following points.

Jeong, J. H., Kug, J. S., Linderholm, H. W., Chen, D., Kim, B. M., & Jun, S. Y. (2014). Intensified Arctic warming under greenhouse warming by vegetation-atmosphere-sea ice interaction. *Environmental Research Letters*, 9(9), 094007.

Main points

1) The authors unnecessarily overstate that vegetation “considerably” enhances Arctic amplification in the abstract (10% is a minor contribution, also in some of the models the remote effects are weak, if not absent, so the authors might want to be slightly more careful). Similarly, at lines 200-201, it would be more fair to say that physiological effects explain some of the intensification (certainly not all of it).

2) The text needs to make a better distinction between on the one hand, causal relationships actually demonstrated by the design of the model experiment, and on the other hand, speculations made by the authors about which processes or feedbacks are responsible for some of the observed effects. For instance, at lines 134-169, the explanation of the effect of vegetation-climate feedbacks or snow-albedo feedbacks represents one possibility. However, because the experiment only investigates radiative versus physiological effects, it cannot definitely disentangle these processes or rigorously attribute causality. For instance, in the case of vegetation feedbacks, one would need an additional experiment with physiological response but fixed vegetation (for the two models that have interactive vegetation). Unlike previous sections, which do rely on the causal relationship investigated by the experiment and are well supported by the figures, this part of the discussion is essentially the authors' interpretation. That is fine, however it should be better acknowledged in the text.

3) The discussion at lines 179-186 over-interprets the results of Supplementary Figure 9. The authors claim that significant relationships exist between the magnitudes of changes in ET, land warming and arctic warming in the inter-model space, however that relationship is not statistically significant according to their figure ($p > 0.05$) and also might be entirely dependent on one of the models being very far from the others. I would strongly recommend revising this section.

4) Including some discussion of important limitations might be helpful, in particular since DGVMs are not able to reproduce current vegetation and albedo dynamics very well in the northern latitudes. For instance, JSBACH has some issues with reproducing a correct albedo in the northern latitudes (e.g. Brovkin et al. 2013), and many models overestimate the mean and trend of LAI compared to observations, particularly over the boreal forest (Murray-Tortarolo et al. 2013). In addition, the role of additional factors potentially limiting plant growth (such as nitrogen availability) is still uncertain and poorly represented. The discussion of the inter-model spread might also be elaborated.

Brovkin, V., Boysen, L., Raddatz, T., Gayler, V., Loew, A., & Claussen, M. (2013). Evaluation of vegetation cover and land-surface albedo in MPI-ESM CMIP5 simulations. *Journal of Advances in Modeling Earth Systems*, 5(1), 48-57.

Murray-Tortarolo, G., Anav, A., Friedlingstein, P., Sitch, S., Piao, S., Zhu, Z., ... & Levis, S. (2013). Evaluation of land surface models in reproducing satellite-derived LAI over the high-latitude Northern Hemisphere. Part I: Uncoupled DGVMs. *Remote Sensing*, 5(10), 4819-4838.

In-text comments

L31-32 : « lead to an increase in plant transpiration » Maybe it could be made clearer at the outset that physiological effects include impacts on both LAI and stomatal conductance which affect transpiration in opposite ways. Otherwise, in the current flow of reading, this sentence reads in contradiction with the abstract (« reduction in evaporative cooling effect and increase in sensible heat flux by physiological effects ») and also with the next paragraph.

L47 : This statement neglects previous global studies on physiological forcing already covering both hydrological aspects and surface temperature (e.g. Swann et al. 2016, Skinner et al. 2018, Lemordant et al. 2018, 2019).

While the introduction explains the notion of stomatal response well, it does not cover the state of the literature on arctic amplification very well. This is important because many alternative mechanisms are under investigation and this contribution mainly aims at contributing to this discussion. For instance, see Dai et al. 2019 and the references therein.

Dai, A., Luo, D., Song, M., & Liu, J. (2019). Arctic amplification is caused by sea-ice loss under increasing CO₂. *Nature communications*, 10(1), 121.

L52: It is unclear what is meant by "two-way interaction", same at L206

L56-61 : This section could be better referenced, notably since those model runs have been used in several previous studies for similar purposes, it's possible to indicate that.

L67-68 : Physiological effects are certainly important, however this should be put in perspective. They are largely offset by the radiative forcing effects causing increased ET (e.g. see Figure 2 in Lemordant et al. 2018 for ET or Skinner et al 2017 Figure 4 for transpiration only). The radiative effects should be at least mentioned in order to avoid misinterpretations of this statement at this stage of the paper. Potentially, delta ET, delta Bowen ratio and delta SAT for the radiative effect could be added as a second column in Figure 1. Already going in this direction, depicting the contours of the radiative forcing effects in Figure 2 was a very nice idea.

L70-71 : It would be nice to add a reference for the increase in LAI and transpiration (or point to a map in the supplement).

L79 : « this area ». It could be useful to locate that specific area in the map (e.g. with a box or an arrow).

L80 : It is not very clear which is the alternative experiment. Is it the phys-piCtI mentioned at line 230 ? Also the SAT effect in the alternative experiment seems about only half as strong, could you comment on that ?

L87 : It is unclear what is meant by the mention « similar to the seasonal cycle of deciduous trees ». Does it mean that such a seasonal pattern is not found for soil evaporation and is entirely due to deciduous trees and not other vegetation ? I would suggest to elaborate or support this statement with appropriate figures.

L95 : How is this number calculated ? over which area, which models, etc.

Figure 3 : The label « delta Land T » is slightly misleading since it represents only 40-70°N as mentioned in the legend. Might be changed to « delta T Land 40-70°N », in consistency with figure 4?

Figure 3 : The nature of DGVM-coupled models with respect to other ESMs is not explained in the manuscript when this figure first appears. This requires an explanation somewhere.

Figure 4, legend: Why is it a five-model mean for the snow concentration? Which models were not taken into account for the average and why?

L98-99: This paragraph ends rather abruptly. It would be nice to either provide an explanation of this remote effect (as done for land warming just above) or simply indicate that this is discussed later in the manuscript.

L106-107: For clarity, consider rephrasing with "... during boreal winter and its magnitude is even comparable ..."

L110: "A previous study", add "A"

L115: "the Pan-Arctic region". It is a bit unclear how this term is used throughout the paper. It seems from the legend in Figure 4 that Pan-Arctic meant >70°N, for some people it means >66°N or even lower, maybe this could be better defined in the introduction.

L123-125: "different physical processes". This could be more specific. The current explanation of the processes leading to a remote and lagged effect is not very convincing here and could be elaborated. This seems like a missing link in the current manuscript.

L125: It is maybe worth mentioning that the remote effects are very strong only in some of the models (e.g. Suppl. Figure 8, bcc-csm-1-1, canESM2, and HadGEM2-ES). Several models show much less propagation to the arctic ocean and little winter effects (e.g. CESM1-BGC, IPSL-CM5A-LR).

L133: consider replacing "to the climate system" with "to arctic warming"

L128-133: This paragraph might potentially be moved to the previous section.

L153-155: For those models with interactive vegetation, replacing bare soils with vegetation or tundra with boreal forests might also change the albedo of the surface, independently of changes in temperature that would melt the snow.

L156-157: This seems like a plausible hypothesis but the presented data cannot really be used to demonstrate it. Maybe this statement should be moderated.

L159: "was mostly consumed in accelerating the melting of sea ice". I find this statement hypothetical and not supported by data. It could be better demonstrated for instance by computing the amount of energy necessary for the additional melting and comparing it with the additional energy actually advected (that might be even done for each model and used to check if there is a consistency, similar to what is attempted in suppl. Figure 9).

I find the absolute reduction in latent heat flux rather intriguing actually, why would it decrease if there is less sea ice, more open water and warmer air? The decrease in latent heat might actually be due to land ET reduction from parcels of land areas $>70^{\circ}\text{N}$, this would be easy to check.

L180-181: Honestly, that relationship is not clear and the regressions are not significant. This statement should be modified.

L183-186: Again that correlation is not significant ($p>5\%$) according to supplementary figure 9 and mostly relies on HadGEM2-ES having such a strong response.

L195 "a previous study" add "a"

L202. I could not find any indication of this in the mentioned reference. Please indicate precisely which models do not include physiological effects.

Response to Reviewers' comments:

Reviewer #1 (Remarks to the Author):

Review of The intensification of Arctic warming as a result of CO₂ physiological forcing by Park et al. for Nature Communications:

This study explores the remote impact of CO₂ physiological forcing on Arctic temperatures. The authors use a suite of CMIP5 models that isolate the impact of CO₂ physiological forcing on projected future climate. They find that elevated CO₂ decreases pan Arctic evapotranspiration via reduced stomatal conductance, which increase local temperatures. Via (unspecified) atmospheric transport mechanisms, this heat is carried to the Arctic where it increases temperatures as well. The heat transport then sets off a number of local feedback mechanisms in the Arctic, including reduced sea ice, increased cloud cover, etc. which further promote Arctic warming. The authors also show that CMIP5 models with DGVMs simulate greater Arctic warming in direct response to the CO₂ physiological forcing (warmer temperatures drive vegetation growth and reduced albedo), and speculate that models without DGVMs may underestimate future Arctic warming.

Although vegetation-Arctic-warming mechanisms have been presented in previous literature, the focus on physiological forcing in this study adds a new wrinkle to our understanding, and I think the topic warrants publication in a journal such as Nature Communications. It will be of interested to a broad audience.

However, I suggest the authors address the following comments before the paper should be considered for publication:

Response: We appreciate the Reviewer #1 for encouraging comments. The reviewer's comments were fully incorporated in the revised manuscript. Our responses to the specific comments are as follows:

Northward Heat Transport Discussion and Supplemental Figure 5:

What are the potential mechanisms for heat transport into the Arctic? Most likely these are atmospheric mechanisms (not ocean).

Additionally, I think there is an error with your energy transport calculation displayed in Supplemental Figure 5. A non-zero transport of energy at the pole is physically impossible. Therefore, a change in the energy transport (as you show) can't be anything but zero.

Additionally, there is a typo in the Figure caption. ϕ in this case is latitude, not longitude.

Response: We really appreciate the reviewer #1 for pointing out very important issue. We found that the non-zero energy transport at the pole is due to the assumption that the climate states are equilibrium conditions for our analysis period. However, the simulations used in this study are a transient run so the time tendency term in the energy budget cannot be ignored. Taking this into consideration, we newly estimated the atmospheric energy convergence into the Arctic basin for the transient conditions using energy budgets and residual methods (Porter et al., 2010; Kay et al., 2012). Following the framework of the previous studies (Nakamura and Oort, 1988; Trenberth et al., 2001), the energy budget of atmospheric column can be denoted as:

$$\frac{\partial E_{ATM}}{\partial t} = NHT_{ATM} + F_{TOA} + F_{SFC}, \quad (1)$$

where $\partial E_{ATM}/\partial t$ is the time change of atmospheric energy storage (W m^{-2}), F_{TOA} is the sum of the net radiation budget at the top of atmosphere (W m^{-2}), F_{SFC} is the net surface energy flux (W m^{-2}) and NHT_{ATM} is the vertically integrated northward heat transport (W m^{-2}). All terms are defined as positive when they increase the atmospheric energy; hence positive downward for the TOA net radiation, positive upward for the net surface energy flux and positive for northward heat transport.

Based on eq. (1), the vertically integrated atmospheric northward heat transport (NHT_{ATM}) can be expressed as:

$$NHT_{ATM} = \frac{\partial E_{ATM}}{\partial t} - F_{TOA} - F_{SFC}. \quad (2)$$

The atmospheric energy storage is written as:

$$E_{ATM} = \frac{1}{g} \int_0^{p_s} (c_p T + k + L_v q + \Phi_s) dp, \quad (3)$$

where p is the pressure (Pa), p_s is the reference surface pressure (hPa), g is the gravitational acceleration (m s^{-2}), c_p is the specific heat of the atmosphere at constant pressure ($\text{J K}^{-1} \text{kg}^{-1}$), T is the temperature (K), k is the kinetic energy (J kg^{-1}), L_v is the latent heat of evaporation (J kg^{-1}), q is the specific humidity (kg kg^{-1}), and Φ_s is the surface geopotential which is not a function of pressure (Trenberth et al., 2001). The contribution of kinetic energy, k , is ignored here due to its comparatively small magnitude (Nakamura and Oort, 1988).

The net radiation at the TOA, F_{TOA} , is defined as:

$$F_{TOA} = F_{SW} - F_{LW}, \quad (4)$$

where F_{SW} is the net shortwave (solar) and F_{LW} is the longwave (thermal) radiation, both in W m^{-2} .

The net surface energy budget at the surface, F_{SFC} , is defined as:

$$F_{SFC} = SW_{SFC} + LW_{SFC} + Q_H + Q_E, \quad (5)$$

where SW_{SFC} and LW_{SFC} are the net surface shortwave and longwave surface radiative fluxes, and Q_H and Q_E are the net surface sensible and latent heat fluxes, all in W m^{-2} .

To support that the atmospheric energy transport is responsible for remotely induced Arctic warming, we calculated the atmospheric northward heat transport (NHT_{ATM}) across the Arctic boundary (at 70°N) using Arctic averaged ($70\text{--}90^\circ\text{N}$) monthly averaged outputs.

The energy, which is transported into the Arctic Ocean, increased during March to July in the final 50 years of CO_2 physiological forcing experiment (Fig. A1). As the Arctic warming is more significant in fall and winter, the poleward energy transport is reversed as consistent with the distribution of vertically integrated temperature changes. In addition, NHT_{ATM} continuously increased during the whole period of simulations due to accumulated CO_2 physiological forcing with increasing CO_2 levels (Fig. A1). These results imply that the NHT_{ATM} plays a role in connecting the extratropical continent warming resulting from the physiological effect to the Arctic warming. We added this figure in the supplementary figure of the revised manuscript.

Figure A1 | Change in atmospheric northward heat transport (NHT_{ATM}) at 70°N resulting from CO₂ physiological forcing. a, Annual cycle of multi-model mean change of atmospheric energy convergence into the Arctic basin. Black line indicates the change in NHT_{ATM} resulting from CO₂ physiological forcing averaged over the final 50 years of the simulations. Black stars indicate significant months for the change in NHT_{ATM} at 90% confidence levels based on a bootstrap method. **b,** Time series of multi-model mean change in atmospheric northward heat transport at 70°N during April– September. NHT_{ATM} was filtered with a 50-year moving average. Light pink shades indicate 90% confidence levels and deep pink shades represent 95% confidence levels based on a bootstrap method.

CO₂ fertilization albedo effects

How does CO₂ fertilization impact the results? You focus on physiological forcing (stomatal conductance), but I expect leaf area index increases from fertilization may also play a role in altering surface albedo and shortwave absorption in all of the models.

Response: As the reviewer pointed out, the difference between the two sets of simulations (denoted *IpctCO2* and *esmFdbk1* in CMIP5, see Supplementary Table 3, 4), includes both physiological effect (stomatal conductance) and CO₂ fertilization effect. These two effects have influences on the plant transpiration in the opposite way: The CO₂ fertilization effect can lead to an increase in plant transpiration by increasing LAI, thereby resulting the cooling effect. The CO₂-induced reduction in the stomatal conductance decreases the transpiration and the resulting enhancement in sensible heat flux directly heat the surface. Despite the offsetting effect from an increase in LAI (Supplementary Fig. 9), a net evapotranspiration decreased under the influence of CO₂ physiological forcing (Fig. 1). These results imply that the fertilization effects have relatively minor role in controlling the transpiration, compared to the physiological effects. These findings are also consistent with the previous studies (Swann et al., 2016; Skinner et al., 2017; Skinner et al., 2018; Hong et al 2018).

In addition to the ET change, we also checked the change in shortwave radiation because an increase in a net shortwave (SW) radiation also contribute to the surface warming resulting from CO₂ physiological forcing. A net SW radiation at the surface could be largely affected by these two factors: An increase in LAI resulting from CO₂ fertilization effect can alter the surface albedo and increase the net SW absorption, thereby contributing to the temperature rise. Reduction of cloud fraction can also be a cause of surface warming because it induces an increase in downward LW radiative flux (Cho et al., 2017).

The change of net SW absorption can be decomposed into:

$$\begin{aligned}\Delta net\ SW\ down &= \Delta(down\ SW - up\ SW) = \Delta(down\ SW \times (1 - \alpha)) \\ &= \Delta down\ SW \times (1 - \alpha) + down\ SW \times (1 - \Delta\alpha) + \Delta down\ SW \times \\ &(1 - \Delta\alpha), (6)\end{aligned}$$

where net SW down ($W\ m^{-2}$) is the net surface shortwave radiative flux, down and up SW ($W\ m^{-2}$) are downward and upward SW radiation at the surface, respectively, and α is the surface albedo (unitless) calculated as:

$$\alpha\ (albedo) = up\ SW / down\ SW. (7)$$

Based on Eq. (6), we can separately quantify the albedo effect and the cloud effect on the total change in net SW radiative flux.

A surface albedo effect, mostly related to the increase in LAI caused by the CO₂ fertilization effect, significantly increases the net surface SW radiation in mid-to-high latitudes especially during boreal spring, autumn and winter (Fig. A2). In summer (June–July–August; JJA), the CO₂ fertilization effect is still significant, but relatively weak. The cloud effect, related to the reduced ET due to the physiological effect, is much greater than the albedo effect, as shown in Fig A2. This result suggests that the increased SW absorption is dominated by the cloud effect, and the albedo change associated with the CO₂ fertilization has only a minor effect, particularly in summer (Figs. A2, A3). The individual models also show the robust cloud effect in high latitude, but the albedo effect is very diverse during summer (Fig. A4).

Figure A2 | Seasonal change in net surface SW radiative flux. a-d, Multi-model mean change of net SW radiation at the surface resulting from CO₂ fertilization-induced albedo effect in MAM (a), JJA (b), SON (c), and DJF (d). e-h, Multi-model mean change of net surface SW radiation by cloud effect resulting from decreased stomatal conductance in MAM (e), JJA (f), SON (g), and DJF (h). Only statistically significant values at the 95% confidence level based on a bootstrap method are shown. (positive: downward)

Figure A3 | Contribution to change in net surface SW absorption in JJA. a,b, Multi-model mean change of net SW radiation. **c-e,** Each term in eq. (6) plotted as a contribution to change of net SW radiative flux from the CO_2 fertilization-induced albedo effect (**c**), cloud effect resulting from decreased stomatal conductance (**d**), and residual effect (**e**). **f,** Sum of changes in net SW absorption based on each term in Eq. (6). Only statistically significant values at the 95% confidence level based on a bootstrap method are shown. (positive: downward)

a Albedo effect: down SW x (1 - Δalbedo)

b Cloud effect: Δdown SW x (1 - albedo)

Figure A4 | Contribution to change in net surface SW radiation in JJA from CMIP5 ESMs. a, Change of net SW radiation at the surface resulting from CO₂ fertilization-induced albedo effect. **b,** Change of net surface SW radiation by cloud effect resulting from decreased stomatal conductance. Only statistically significant values at the 95% confidence level based on a bootstrap method are shown. (positive: downward)

Typos/Grammar

There are a number of grammatical mistakes in the paper, including in the Abstract.

Response: We have carefully reexamined the manuscript to improve the English.

References

Cho, M. H. *et al.* Vegetation-cloud feedbacks to future vegetation changes in the Arctic regions. *Clim. Dyn.* **50**, 3745–3755 (2018).

Hong T. *et al.* The response of vegetation to rising CO₂ concentrations plays an important role in future changes in the hydrological cycle. *Theor. Appl. Climatol.* **136**, 135–144 (2019).

Kay J. E. *et al.* The Influence of Local Feedbacks and Northward Heat Transport on the Equilibrium Arctic Climate Response to Increased Greenhouse Gas Forcing. *J. Clim.* **25**, 5433–5450 (2012).

Nakamura N. & Oort A. H. Atmospheric heat budgets of the polar regions. *J. Geophys. Res.* **93**, 9510–9524 (1988).

Porter D. F., Cassano J. J., Serreze M. C. & Kindig D. N. New estimates of the large-scale Arctic atmospheric energy budget. *J. Geophys. Res.* **115**, D08108 (2010).

Skinner, C. B., Poulsen, C. J., Chadwick, R., Diffenbaugh, N. S. & Fiorella, R. P. The role of plant CO₂ physiological forcing in shaping future daily-scale precipitation. *J. Clim.* **30**, 2319–2340 (2017).

Skinner, C. B., Poulsen, C. J. & Mankin, J. S. Amplification of heat extremes by plant CO₂ physiological forcing. *Nat. Commun.* **9**, 1094 (2018).

Swann, A. L. S., Hoffman, F. M., Koven, C. D. & Randerson, J. T. Plant responses to increasing CO₂ reduce estimates of climate impacts on drought severity. *Proc. Natl. Acad. Sci. USA* **113**, 10019–10024 (2016).

Trenberth K. E., Caron J. M. & Stepaniak D. P. The atmospheric energy budget and implications for surface fluxes and ocean heat transports. *Clim. Dyn.* **17**, 259–276 (2001).

Reviewer #2 (Remarks to the Author):

Review for NCOMMS-19-31397 submission from Park et al. entitled “The intensification of Arctic warming as a result of CO₂ physiological forcing.”

Park and colleagues leverage the 8 models participating in the C4MIP set of CMIP5-era simulations to show that physiological forcing from CO₂ reduces boreal ET, leading to a partitioning of energy towards sensible heat, thereby amplifying Arctic warming over the land, and in particular, leads to remote warming of surface air over the ocean. The premise of the manuscript is that the remote forcing from stomatal responses is under-explored and the Arctic is a region potentially sensitive to such forcing.

I have a few comments that I hope the authors will find helpful as they prepare a revision for this or another journal.

Response: We thank the reviewer #2 for insightful and critical comments. The reviewer’s comments were fully incorporated in the revised manuscript. Particularly, we made an effort to clarify the mechanism for the remote forcing of the physiological effect, and their robustness. Our responses to the specific comments are as follows:

1. Generalizability and mechanisms: The premise of the manuscript and its claims to novelty rest on the remote forcing of Arctic temperatures by boreal forest physiological responses to CO₂ and the presentation of a mechanism by which the forcing occurs. The problem is that it is not clear that the remote forcing exists anywhere but in the ensemble mean, nor have the authors actually provided a physical mechanism by which this remote forcing takes place that explains the variability across the models. Much of the basis of the remote forcing claim rests on the results presented in Figs. 2 and S5, which show a time-latitude change in ET and temperature due to Phy and Rad, and the total northward energy transport in summer and annually, respectively. Fig. S6 also contributes to this picture. While it’s clear the physiological forcing generates conditions for Arctic warming in the ensemble mean, it’s not clear to me that the authors have identified the mechanism, nor is it clear whether such mechanisms exist on a model-by-model basis. Looking at some of the supplementary figures, it appears that a lot of the ensemble mean response over the Arctic is driven by HadGEM2-ES and bcc-csm1-1 in fall and winter (boreal). So it’s not clear that the authors are explaining the actual process by which this forcing occurs. For example, Fig S9b shows that the HadGEM2-ES model really drives the remote forcing seen in the ensemble mean plots presented in the main (and it has very little vegetation changes over the regions that purportedly drive the remote forcing, Fig. S13). It would be helpful would be to see the individual energy transport changes for each model (Fig. S5), and to see the changes in cloud fractions at seasonal scales for each model as they present for temperature and ET in the suppmat. Where are the error bars on the bar plot presented in Fig. 4e and f for example? Connecting the ET response to energy partitioning to remote warming (and cloud fractions) across models would actually address this. As of now, I’m not entirely convinced that the remote forcing is a generalized and physically-consistent response within the models due to Phy forcing—it seems to be an outlier story the authors are projecting onto all models.

Response: Thanks for suggesting good points. As the reviewer pointed out, it is important to know whether or not the ensemble mean is controlled by a few outlier models. As the reviewer suggested, we calculated the physiological effect after the HadGEM2-ES, having the largest responses, is excluded, in order to examine the dependency of MME results on the single model. Excluding HadGEM2-ES, we reproduced all main Figures in the manuscript (Figs. B1,B2,B3

and B4). Despite the exclusion of model having the largest response, those Figures show very similar results to the original Figures though the magnitude is a bit altered. We also checked the atmospheric northward heat transport with and without HadGEM2-ES, but still the results are quite similar (Figs. B5,B6). These results suggest that our main arguments are robust and are not sensitive to the model selection.

Supplementary Fig. 11 shows the individual model's changes in the land ET and temperature, and Arctic temperature. Except only one case (the change of ET from GFDL-ESM2M, see Supplementary Notes), all responses to the physiological forcing from each individual model are at least the same sign, suggesting the inter-model consistency and the quite robust response to physiological effects. Of course, there is a considerable inter-model diversity in terms of their magnitude. Even so, the most models are simulating the major processes that the physiological effect causes the reduction of ET and the resulting land surface warming, which leads to the increase in northward energy transport in boreal summer, which triggers the Arctic climate feedback, and eventually induces enhanced Arctic amplification. While the most models simulate consistently main processes on the physiological process, we found that the models' diversity in terms of the magnitude strongly depends on the strength of the local feedbacks (i.e. albedo feedback and cloud feedback) in the continent and the Arctic as discussed in the main text (also Fig. B7). In particular, we found that HadGEM2-ES has the strongest local feedback related to sea ice in the Arctic Ocean.

Figure B1 | Change in the annual mean evapotranspiration, Bowen ratio and near-surface air temperature resulting from CO₂ physiological forcing excluding HadGEM2-ES. a–c, Multi-model mean change in the annual mean evapotranspiration (a), Bowen ratio (sensible heat flux/latent heat flux) (b), and near-surface air temperature (c) resulting from CO₂ physiological forcing excluding HadGEM2-ES. Only significant values at the 95% confidence level based on a bootstrap method are shown.

Figure B2 | Impacts of the physiological forcing on the evapotranspiration and surface air temperature excluding HadGEM2-ES. a,b, Zonally and monthly averaged change in the evapotranspiration (a) and surface air temperature (b) excluding HadGEM2-ES. The shading represents the change resulting from CO₂ physiological forcing. The contouring represents the change resulting from CO₂ radiative forcing. The contour intervals for radiative forcing are 0.1 mm day⁻¹ in a and 1.5 K in b.

Figure B3 | The ratio (percentage) between changes in the surface air temperature in response to CO₂ physiological forcing and CO₂ radiative forcing excluding HadGEM2-ES. Ratio of the change in the near-surface air temperature resulting from CO₂ physiological effects to that resulting from CO₂ radiative forcing ($[\text{CO}_2 \text{ physiological forcing} / \text{CO}_2 \text{ radiative forcing}] \times 100$) in the continents (40°–70°N) and Arctic region (70°–90°N), respectively. Each bar shows the area-weighted average of multi-model ensemble excluding HadGEM2-

ES. The black dots represent the individual results from ESMs excluding HadGEM2-ES. The error bar for each column indicates the range of the 95% confidence level on the basis of a bootstrap method.

Figure B4 | Change in the annual mean snow, sea ice, cloud and surface albedo resulting from CO₂ physiological forcing excluding HadGEM2-ES. a–d, Multi-model mean change of the annual mean snow concentration (a), sea ice concentration (b), total cloud fraction (c) and surface albedo (d) excluding HadGEM2-ES. Only significant values at the 95% confidence level based on a bootstrap method are shown. e–f, Annual cycle of change in the surface air temperature (red bar) resulting from CO₂ physiological effect with the ratio between change in the snow concentration (blue line) in response to physiological forcing and radiative forcing in landmasses (40°–70°N) excluding HadGEM2-ES (e). The same as in e, but in the Arctic region (70°–90°N), with the ratio between change in the sea ice concentration (blue line) in response to physiological forcing and radiative forcing excluding HadGEM2-ES (f). The error bars represent a range of the 95% confidence level on the basis of a bootstrap method. All of the values are area-weighted averages of seven ESMs excluding HadGEM2-ES, except for the snow concentration, which is an average of five ESMs because GFDL-ESM2M, HadGEM2-ES, and IPSL-CM5A-LR do not provide the surface snow area fraction.

Figure B5 | Change in atmospheric northward heat transport (NHT_{ATM}) at 70°N resulting from CO₂ physiological forcing. a, Annual cycle of multi-model mean change of atmospheric energy convergence into the Arctic basin. Black line indicates the change in NHT_{ATM} resulting from CO₂ physiological forcing averaged over the final 50 years of the simulations. Black stars indicate significant months for the change in NHT_{ATM} at 90% confidence levels based on a bootstrap method. **b,** Time series of multi-model mean change in atmospheric northward heat transport at 70°N during April–September. NHT_{ATM} was filtered with a 50-year moving average. Light pink shades indicate 90% confidence levels and deep pink shades represent 95% confidence levels based on a bootstrap method.

Figure B6 | Change in atmospheric northward heat transport (NHT_{ATM}) at 70°N resulting from CO₂ physiological forcing excluding HadGEM2-ES. a, Annual cycle of multi-model mean change of atmospheric energy convergence into the Arctic basin excluding HadGEM2-ES. Black line indicates the change in NHT_{ATM} resulting from CO₂ physiological forcing averaged over the final 50 years of the simulations. Black stars indicate significant months for the change in NHT_{ATM} at 90% confidence levels based on a bootstrap method. **b,** Time series of multi-model mean change in atmospheric northward heat transport at 70°N during April–September excluding HadGEM2-ES. NHT_{ATM} was filtered with a 50-year moving average. Light pink shades indicate 90% confidence levels and deep pink shades represent 95% confidence levels based on a bootstrap method.

Figure B7 | Impacts of local feedbacks resulting from CO₂ physiological forcing on the surface warming in the continent (40°–70°N) and the Arctic (70°–90°N). **a**, Scatterplot of change in the total cloud fraction versus near-surface air temperature over continent (40°–70°N) resulting from CO₂ physiological forcing during summer. **b**, Scatterplot of annual mean change in the sea ice fraction versus near-surface air temperature over Arctic region (70°–90°N) resulting from CO₂ physiological forcing. All values are area-weighted averages of CMIP5 multi-model ensemble.

As the reviewer suggested, we added the error bars in Figs 4e and f, which show quite significant results in all months. Additionally, we added the figure of change in the cloud fraction at seasonal scales for each individual model in Supplementary Fig. 10.

2. Context relative to Rad-only and Total effects. Some of the work by the authors does present the relative contributions of physiological forcing relative to the Rad-only (e.g., Fig. 2, and 4, etc), the effect size we're talking about here can be quite small, but this reality remains under-emphasized in the manuscript as written. So if there is remote forcing of Arctic, it would be helpful (and interesting) to know if this remote forcing has a nonlinear interaction with the warming from the Rad-only estimate. That is instead of $\text{Phy} + \text{Rad} = \text{Total}$, as assumed in the current form, could the total effects actually be: $\text{Phy} + \text{Rad} + e = \text{Total}$, implying that the total effects can't be linearly decomposed as the authors have done. Something to show that the linearity assumption holds, and that puts the actual Phy effects on Arctic amplification in context fractionally, would be good. It would be really interesting to know if the Phy and Rad are interactive (and nonlinear).

Response: We thank the reviewer #2's interesting suggestion. We totally agree that a nonlinear interaction (e) between physiological forcing (Phy) and radiative forcing (Rad) exists. In other words, the global warming signal by the radiation forcing plays a role in enhancing the physiological effect.

To quantify this effect, we compared two different physiological effects. One is defined as a difference between the total and Rad-only simulations. The other is defined as a difference between Phy-only and control simulations. The former includes the nonlinear or synergy

effect. Though the surface warming decreases in the magnitude, the only physiological forcing excluding a nonlinear interaction provides the consistent results, showing significant Arctic warming, which confirms the robustness of our findings (Fig. B8). When excluding an interaction, the magnitude of annual mean change of Arctic warming decreased by 24% (Table B1). This nonlinear effect generally causes the additional surface warming, especially in Arctic region during boreal winter (Table B1). It is possible that the positive feedback in Arctic Ocean regime might have been amplified due to this additional temperature rise resulting from an interaction between Phy and Rad, but further analysis is needed to confirm this hypothesis.

Figure B8 | Change in the annual mean evapotranspiration, Bowen ratio and near-surface air temperature resulting from alternative CO₂ physiological forcing excluding a synergy effect. a–c, Multi-model mean change of the annual mean evapotranspiration (a), Bowen ratio (sensible heat flux/latent heat flux) (b), and near-surface air temperature (c) resulting from only CO₂ physiological forcing excluding a nonlinear interaction between the physiological forcing and radiative forcing. Only significant values at the 95% confidence level based on a bootstrap method are shown.

Table B1 | Change in temperature caused by only-CO₂ physiological forcing (Phy-only) and by a nonlinear interaction between physiological forcing and radiative forcing (e).

	ANN	MAM	JJA	SON	DJF
Mid-to-high latitude continents (40°–70°N)					
Phy-only	0.57	0.49	0.87	0.55	0.39
e	0.15	0.10	0.18	0.09	0.22
Arctic region (70°–90°N)					
Phy-only	0.57	0.43	0.31	0.92	0.62
e	0.18	0.21	0.21	-0.08	0.37

3. The authors claim at line 70 that the ET drop suggests that fertilization effects are smaller than stomatal ones on total ET. This may be true in the Phy-only sense, but the authors have not actually tested it. It is not true when you consider the longer and warmer growing seasons due to Rad interacting with Phy, which leads to a general increase in high-latitude partitioning of precipitation to vegetation and often an increase in ET. So the authors could make sure that is clearer, because they don't actually perform a model-by-model analysis of LAI-driven v. stomatal-driven ET changes. Also, the response as presented in Fig. 1 is really only significant for extremely high latitude vegetation, which is going to be a function of the PFTs the models place there. For example, how much of the ensemble mean ET response is driven by the ET in the IPSL model in summer (Fig. S7b)? If that model were dropped, could the authors still make the claim presented in line 70?

Response: As the reviewer suggested, we re-calculated Fig. 1 by excluding the IPSL-CM5A-LR model (See Fig. B9). The significant area for change in the evapotranspiration is slightly reduced, but still the overall pattern is very similar to the original Figure. More importantly, the Bowen ratio is increased over the most extratropical continent region in NH.

We agree with the reviewer that we cannot exactly separate the response of LAI-driven and stomatal driven ET changes. However, we know that their signs are mostly opposite, so that the reduced ET suggests that the stomatal-driven ET changes are greater than the LAI-driven ET. Also, many previous literatures already pointed out this argument (Swann et al., 2016; Skinner et al., 2017; Skinner et al., 2018; Hong et al 2018). In particular, Skinner et al. 2017 clearly showed that change in ET from CO₂_Stomata is more dominant than CO₂_LAI through experiments separating these two effects using CCSM4 (Figures 2,4 in Skinner et al 2017). In addition, we checked their relative contribution to the change in the downward shortwave (SW) radiation since the LAI-related and stomatal-related effects have an influence on the solar radiation in different ways. That is, the LAI-related effect changes the SW absorption by altering the surface albedo, and the stomatal-related effect is somewhat more related to the increases in the downward solar radiation due to the reduction of total cloud fraction. The change of the net SW radiation can be separated into three terms as follows:

$$\Delta_{net\ SW\ down} = \Delta_{down\ SW} \times (1 - \alpha) + down\ SW \times (1 - \Delta\alpha) + \Delta_{down\ SW} \times (1 - \Delta\alpha) \quad (1)$$

As shown in Fig. B10, the cloud effect is greater than the albedo effect during boreal summer.

Overall, these results suggest that the impact of stomatal closure has a predominant role in the surface warming under elevated CO₂ levels. However, corresponding to the reviewer’s comment, we re-write this part with more careful argument as follows:

(P5 L80-84) “The fertilization effect plays a role in increasing ET due to the resulting increased LAI (Supplementary Fig. 9) opposite to the effect of the stomatal closure^{5,10,37}. Taking this into consideration, this overall decrease in ET suggests that the effect of stomatal closure is greater than the CO₂ fertilization effects on evapotranspiration under elevated CO₂, which is consistent with the previous studies^{12,31,34,37}.”

Figure B9 | Change in the annual mean evapotranspiration, Bowen ratio and near-surface air temperature resulting from CO₂ physiological forcing excluding IPSL-CM5A-LR. a–c, Multi-model mean change in the annual mean evapotranspiration (a), Bowen ratio (sensible heat flux/latent heat flux) (b), and near-surface air temperature (c)

resulting from CO₂ physiological forcing excluding IPSL-CM5A-LR. Only significant values at the 95% confidence level based on a bootstrap method are shown.

Figure B10 | Contribution to change in net surface SW absorption in JJA. a,b, Multi-model mean change of net SW radiation. c-e, Each term in eq. (1) plotted as a contribution to change of net SW radiative flux from the CO₂ fertilization-induced albedo effect (c), cloud effect resulting from decreased stomatal conductance (d), and residual effect (e). f, Sum of changes in net SW absorption based on each term in Eq. (1). Only statistically significant values at the 95% confidence level based on a bootstrap method are shown. (positive: downward)

References

- Hong T. *et al.* The response of vegetation to rising CO₂ concentrations plays an important role in future changes in the hydrological cycle. *Theor. Appl. Climatol.* **136**, 135–144 (2019).
- Skinner, C. B., Poulsen, C. J., Chadwick, R., Diffenbaugh, N. S. & Fiorella, R. P. The role of plant CO₂ physiological forcing in shaping future daily-scale precipitation. *J. Clim.* **30**, 2319–2340 (2017).
- Skinner, C. B., Poulsen, C. J. & Mankin, J. S. Amplification of heat extremes by plant CO₂ physiological forcing. *Nat. Commun.* **9**, 1094 (2018).

Swann, A. L. S., Hoffman, F. M., Koven, C. D. & Randerson, J. T. Plant responses to increasing CO₂ reduce estimates of climate impacts on drought severity. *Proc. Natl. Acad. Sci. USA* **113**, 10019–10024 (2016).

Reviewer #3 (Remarks to the Author):

Summary

This study investigates the role of plant physiological response in the context of rising atmospheric CO₂ for arctic warming amplification. The results are based on a set of CMIP5 climate runs designed to disentangle atmospheric effects (i.e. the radiative global response to elevated CO₂) from the physiological effects (i.e. the vegetation response to elevated CO₂ and its local effect on climate), following an approach very similar to previous studies (e.g. Swann et al. 2016, Skinner et al. 2018, Lemordant et al. 2018, 2019).

In line with previous findings, the authors find that increased plant water use efficiency in boreal regions leads to a net decrease in plant transpiration, thus offsetting an overall increase in LAI (and the associated increase in water demand). This effect is almost exclusively occurring during summer and leads to increased sensible heat, reduced cloud cover, higher net radiation and overall higher surface air temperatures over land. The authors then argue that this additional energy is advected above the arctic ocean, leading to a decrease of sea ice cover and increased cloud fraction, and explaining a delayed winter-time warming over this area revealed by the experiment. The authors calculate that the physiological contribution represents on average 10% of the radiative effect, and thereby positively contributes to the so-called Arctic warming amplification.

Lemordant, L., Gentine, P., Swann, A. S., Cook, B. I., & Scheff, J. (2018). Critical impact of vegetation physiology on the continental hydrologic cycle in response to increasing CO₂. *Proceedings of the National Academy of Sciences*, 115(16), 4093-4098.

Swann, A. L., Hoffman, F. M., Koven, C. D., & Randerson, J. T. (2016). Plant responses to increasing CO₂ reduce estimates of climate impacts on drought severity. *Proceedings of the National Academy of Sciences*, 113(36), 10019-10024.

Skinner, C. B., Poulsen, C. J., & Mankin, J. S. (2018). Amplification of heat extremes by plant CO₂ physiological forcing. *Nature communications*, 9(1), 1094.

Lemordant, L., & Gentine, P. (2019). Vegetation Response to Rising CO₂ Impacts Extreme Temperatures. *Geophysical Research Letters*, 46(3), 1383-1392.

Overall assessment

The CO₂ physiological effects on evaporative demand and their consequences for LAI, surface energy fluxes and air temperature as diagnosed from CMIP5 experiments are relatively well covered by the existing literature. What essentially distinguishes this study from previous work is its focus on the arctic region and its contribution to the scientific discussion on arctic warming amplification. This study is in the line of previous work by some of the coauthors (e.g. Jeong et al. 2014) and demonstrates that physiological effects have a modest (10%) but measurable contribution to arctic warming (independently of changes in vegetation type or cover), thus bringing an incremental but new contribution to our understanding of arctic amplification. The authors make a convincing case, provide good supporting information and the study seems scientifically sound as far as I can judge, with the exception of some points. My main criticism relates to the following points.

Jeong, J. H., Kug, J. S., Linderholm, H. W., Chen, D., Kim, B. M., & Jun, S. Y. (2014).

Intensified Arctic warming under greenhouse warming by vegetation–atmosphere–sea ice interaction. *Environmental Research Letters*, 9(9), 094007.

Response: We thank the reviewer #3 for his/her helpful comments. The reviewer's comments were fully incorporated in the revised manuscript. Our responses to the specific comments are as follows:

Main points

1) The authors unnecessarily overstate that vegetation “considerably” enhances Arctic amplification in the abstract (10% is a minor contribution, also in some of the models the remote effects are weak, if not absent, so the authors might want to be slightly more careful). Similarly, at lines 200-201, it would be more fair to say that physiological effects explain some of the intensification (certainly not all of it).

Response: We have deleted the word “considerably” to avoid the overstatement as follows: (P2 L19-21) “Here we show that the presence of vegetation at high latitudes enhances the Arctic amplification *via* remote and time-delayed physiological forcing processes.”

(P11 L223-225) “These findings suggest that the physiological effects can somewhat explain the intensification of Arctic warming at a high level of CO₂ concentration in the future.”

2) The text needs to make a better distinction between on the one hand, causal relationships actually demonstrated by the design of the model experiment, and on the other hand, speculations made by the authors about which processes or feedbacks are responsible for some of the observed effects.

For instance, at lines 134-169, the explanation of the effect of vegetation-climate feedbacks or snow-albedo feedbacks represents one possibility. However, because the experiment only investigates radiative versus physiological effects, it cannot definitely disentangle these processes or rigorously attribute causality. For instance, in the case of vegetation feedbacks, one would need an additional experiment with physiological response but fixed vegetation (for the two models that have interactive vegetation). Unlike previous sections, which do rely on the causal relationship investigated by the experiment and are well supported by the figures, this part of the discussion is essentially the authors' interpretation. That is fine, however it should be better acknowledged in the text.

Response: As the reviewer suggested, we describe our interpretation part more carefully as follow:

(P9-10 L169-208) “The surface warming at mid-to-high latitudes during summer is greatest in HadGEM2-ES and MPI-ESM-LR (Supplementary Figs. 8,11). Interestingly, an increase of net SW absorption in JJA is greatest in these two models (Supplementary Table 7). In this experimental design, the net SW absorption can be largely affected by these two factors: An increase in LAI resulting from CO₂ fertilization effect can alter the surface albedo and increase the net SW absorption, thereby contributing to the temperature rise. Reduction of cloud fraction can also be a cause of surface warming because it induces an increase in downward LW radiative flux⁴⁰. From their relative contributions, we found that vegetation-cloud feedback has a dominant role in the increased net SW absorption during summer, thereby contributing the continental warming (40°–70°N) particularly in summer (Supplementary Fig. 13). As a result, HadGEM2-ES and MPI-ESM-LR show the greatest warming in JJA due to this greatest cloud effect despite the moderate reduction of ET (Supplementary Fig. 10). In addition, the relative magnitude of the vegetation-cloud feedback in ESMs seems to explain the inter-model diversity of the land surface warming (40°–70°N) in JJA ($r=-0.79$, $P=0.02$) (Supplementary Fig. 12a).

Another local feedback might be triggered by physiological forcing over the continents (40°–70°N). As shown in Fig. 4, a snow concentration and a surface albedo in high latitudes significantly declined in response to the CO₂ physiological forcing. The warming resulting from physiological effects presumably melts snow and the resulting less-snow covered surface absorbs more solar radiation (Supplementary Table 5). Furthermore, an increase of LAI from the fertilization effects and the land cover change in models with interactive vegetation might partially contribute to the surface warming by altering the surface albedo independently of a change in temperature and would melt the snow as noticed previously^{45,46} (Supplementary Figs. 9,14). Consequently, this snow–albedo feedback may help enhance and maintain the land surface warming throughout the year, especially in high latitudes where the surface albedo is relatively high due to the high snow cover (Fig. 4e).

In contrast to the change of cloud cover over the continents (40°–70°N), the cloud formation is enhanced in the Arctic region especially during winter (Fig. 4 and Supplementary Table 6). This increased cloud fraction additionally intensifies the surface warming by decreasing the outgoing longwave radiation especially in non-summer season^{47,48} (Supplementary Table 6). Although it is difficult to prove the causality in this experiment, it is conceived that this increase in cloud formation contributes to the Arctic sea ice loss, which in turn causes the increase in water vapor from the newly opened Arctic waters, as proposed previously^{21,48} (Fig. 4). In summary, the cloud feedback in the Arctic can enhance the surface warming by increasing a downward LW radiation, and in turn, the enhanced surface warming can accelerate the sea-ice loss, thereby causing positive feedback during the cold season. On the whole, our results suggest that the local feedbacks triggered by physiological effects might additionally contribute to the amplified and maintained surface warming in both continents and the Arctic Ocean.”

3) The discussion at lines 179-186 over-interprets the results of Supplementary Figure 9. The authors claim that significant relationships exist between the magnitudes of changes in ET, land warming and arctic warming in the inter-model space, however that relationship is not statistically significant according to their figure ($p > 0.05$) and also might be entirely dependent on one of the models being very far from the others. I would strongly recommend revising this section.

Response: Due to the small number of the models, it is difficult to have significant relation even though the correlation is not low. As the review suggested, we removed the part on the inter-model relationship. Also, we have revised this paragraph as follows:

(P11 L211-221) “So far, it has been shown based on a multi-model mean that distinctive Arctic warming occurs due to the physiological effects, but this conclusion can be model-dependent because the structure of models and the parameterization schemes are different from each other. The magnitudes and spatial patterns of change in ET and temperature were found to be diverse (Supplementary Figs. 7,8). Nevertheless, scatterplot, which represent the ET and temperature responses from the individual model, shows that most models consistently simulate the reduction in ET and the resulting surface warming over the continents (Supplementary Fig. 11). In addition, most of the models simulate enhanced Arctic warming as a result of physiological effects. Since most models consistently simulate the continent and Arctic warming in spite of the different magnitude, our results are not sensitive to the subsampling of the models.”

4) Including some discussion of important limitations might be helpful, in particular since DGVMs are not able to reproduce current vegetation and albedo dynamics very well in the

northern latitudes. For instance, JSBACH has some issues with reproducing a correct albedo in the northern latitudes (e.g. Brovkin et al. 2013), and many models overestimate the mean and trend of LAI compared to observations, particularly over the boreal forest (Murray-Tortarolo et al. 2013). In addition, the role of additional factors potentially limiting plant growth (such as nitrogen availability) is still uncertain and poorly represented. The discussion of the inter-model spread might also be elaborated.

Brovkin, V., Boysen, L., Raddatz, T., Gayler, V., Loew, A., & Claussen, M. (2013). Evaluation of vegetation cover and land-surface albedo in MPI-ESM CMIP5 simulations. *Journal of Advances in Modeling Earth Systems*, 5(1), 48-57.

Murray-Tortarolo, G., Anav, A., Friedlingstein, P., Sitch, S., Piao, S., Zhu, Z., ... & Levis, S. (2013). Evaluation of land surface models in reproducing satellite-derived LAI over the high-latitude Northern Hemisphere. Part I: Uncoupled DGVMs. *Remote Sensing*, 5(10), 4819-4838. Response: We have briefly mentioned this in the revised manuscript and then discussed the DGVM's limitations in the revised Supplementary notes in detail as follows: (P11-12 L232-234) "There are still the limitations of land surface models in simulating current LAI and the albedo dynamics (see the Supplementary Notes)."

(Supplementary Information, P3 L39-56) "GFDL-ESM2M, HadGEM2-ES and MPI-ESM-LR were coupled with dynamic vegetation modules (Supplementary Table 2), which simulate shifts in the land cover and its associated biogeochemical and hydrological processes as a response to climate change. DGVMs are currently the best available ways to represent vegetation dynamics used in global scale studies and also the coupling of DGVMs and GCMs provide an opportunity to assess the vegetation-atmosphere interactions in climate simulations⁷. However, several studies have consistently suggested that dynamic vegetation models have serious weaknesses in reproducing the observed vegetation and albedo dynamics in the Northern hemisphere and thereby contribute to bias in climate simulations⁸⁻¹⁰. The vegetation models overestimate the mean and trend of LAI particularly in the boreal forest and also simulate the longer growing season compared to the observations⁸⁻¹⁰. In addition, nutrients cycles, such as nitrogen that limits the capacity of the plants growth and is tightly coupled with the carbon cycle, are not fully integrated and poorly represented in DGVMs⁷. Nevertheless, the incorporation of dynamic vegetation modules appears to have little effect on intermodel differences of projection in LAI¹¹. There also remain the issues that the impact of the dynamic vegetation modules on a coupled DGVM-GCM experiment depends on the strength of the land-atmosphere coupling and this coupling strength can vary widely between models⁷."

In-text comments

L31-32 : « lead to an increase in plant transpiration » Maybe it could be made clearer at the outset that physiological effects include impacts on both LAI and stomatal conductance which affect transpiration in opposite ways. Otherwise, in the current flow of reading, this sentence reads in contradiction with the abstract (« reduction in evaporative cooling effect and increase in sensible heat flux by physiological effects ») and also with the next paragraph.

Response: We have clarified it at the outset of introduction as follows:

(P3 L28-29) "Physiological responses to increasing CO₂ include changes in leaf area index (LAI) and stomatal conductance, and those affect the plant transpiration in opposite ways."

L47 : This statement neglects previous global studies on physiological forcing already covering

both hydrological aspects and surface temperature (e.g. Swann et al. 2016, Skinner et al. 2018, Lemordant et al. 2018, 2019).

Response: We have revised that sentence as follows:

(P3 L46-49) “This physiological effect has a potential to remotely alter the entire climate system through the redistribution of the surface energy and disturbance of hydrological cycle, but still, the remote impacts of physiological effect on the climate system are unclear especially in the Arctic region (north of 70°N).”

While the introduction explains the notion of stomatal response well, it does not cover the state of the literature on arctic amplification very well. This is important because many alternative mechanisms are under investigation and this contribution mainly aims at contributing to this discussion. For instance, see Dai et al. 2019 and the references therein.

Dai, A., Luo, D., Song, M., & Liu, J. (2019). Arctic amplification is caused by sea-ice loss under increasing CO₂. *Nature communications*, 10(1), 121.

Response: We have covered the literature on Arctic amplification in the revised manuscript as follows:

(P4 L51-57) “Many mechanisms have been suggested to explain the Arctic amplification including a role of diminishing sea-ice^{17,18}, seasonal storage and release of the absorbed shortwave (SW) radiation coupling with sea ice loss¹⁹⁻²¹, enhanced downward longwave (LW) radiation due to an increase in water vapor and cloud fraction^{22,23}, ocean biogeochemical feedback^{24,25}, increased poleward energy transport^{26,27} and other processes^{28,29}. However, their relative contributions are still under debate and also many alternative mechanisms are under investigation.”

L52: It is unclear what is meant by “two-way interaction”, same at L206

Response: We have deleted that word.

L56-61 : This section could be better referenced, notably since those model runs have been used in several previous studies for similar purposes, it’s possible to indicate that.

Response: We cited the previous studies that use the same experimental simulations.

(P4 L65-69) “In line with previous studies^{12,13,31-34}, we respectively quantified the physiological forcing (Phy), which includes the CO₂ fertilization effect and the dependency of stomatal conductance on CO₂, and CO₂ radiative forcing (Rad) (average CO₂ concentrations ~823 ppm) using carbon-climate feedback experiments (see the Methods section and Supplementary Tables 3,4).”

L67-68 : Physiological effects are certainly important, however this should be put in perspective. They are largely offset by the radiative forcing effects causing increased ET (e.g. see Figure 2 in Lemordant et al. 2018 for ET or Skinner et al 2017 Figure 4 for transpiration only). The radiative effects should be at least mentioned in order to avoid misinterpretations of this statement at this stage of the paper. Potentially, delta ET, delta Bowen ratio and delta SAT for the radiative effect could be added as a second column in Figure 1. Already going in this direction, depicting the contours of the radiative forcing effects in Figure 2 was a very nice idea.

Response: We have added the supplementary figure for CO₂ radiative forcing (Supplementary Fig. 1) and also mentioned the role of CO₂ radiative effects on transpiration to avoid the misinterpretations as follows:

(P5 L75-79) “In contrast with the radiative effect inducing the increase in ET due to enhanced water-demand from the temperature rise (Supplementary Fig. 1), physiological effects cause a

conspicuous and significant reduction in the annual mean ET in densely vegetated areas of the tropics and mid-to-high latitudes (Fig. 1a) in line with previous studies^{12,31,32,34-37}.”

L70-71 : It would be nice to add a reference for the increase in LAI and transpiration (or point to a map in the supplement).

Response: We have added the figure of changes in LAI at seasonal scales for each model (Supplementary Fig. 9) and also referenced the literature on the increase in transpiration due to increased LAI.

(P5 L79-80) “The fertilization effect plays a role in increasing ET due to the resulting increased LAI (Supplementary Fig. 9) opposite to the effect of the stomatal closure^{5,10,37}.”

L79 : « this area ». It could be useful to locate that specific area in the map (e.g. with a box or an arrow).

Response: This area indicates “Arctic Ocean”. We have replaced the word “this area” to “Arctic Ocean”.

L80 : It is not very clear which is the alternative experiment. Is it the phys-piCtl mentioned at line 230 ? Also the SAT effect in the alternative experiment seems about only half as strong, could you comment on that ?

Response: Yes, it is the Phys-piCtl. It is shown that the radiation effect can amplify the physiological effect. Previous studies mentioned it as a synergy effect, or nonlinear interaction (Bathiany et al. 2014; Skinner et al. 2017)

L87 : It is unclear what is meant by the mention « similar to the seasonal cycle of deciduous trees ». Does it mean that such a seasonal pattern is not found for soil evaporation and is entirely due to deciduous trees and not other vegetation? I would suggest to elaborate or support this statement with appropriate figures.

Response: We wanted to say that the physiological effects (stomatal conductance) increase from spring, the maximum occurs in summer and after then physiological effects gradually decrease similarly with the seasonal cycle of deciduous trees. We have deleted the phrase “similar to the seasonal cycle of deciduous trees” in the revised manuscript to avoid the misinterpretation.

L95 : How is this number calculated ? over which area, which models, etc.

Response: This value is the area-weighted average of the downward solar radiation over landmasses (40°–70°N) in summer (JJA) from CMIP5 multi-model ensemble. It is the same as the value in Supplementary Table 5.

Figure 3 : The label « delta Land T » is slightly misleading since it represents only 40-70°N as mentioned in the legend. Might be changed to « delta T Land 40-70°N », in consistency with figure 4?

Response: corrected.

Figure 3 : The nature of DGVM-coupled models with respect to other ESMs is not explained in the manuscript when this figure first appears. This requires an explanation somewhere.

Response: We have modified the Figure 3 in order not to emphasize the nature of DGVM-coupled models in the revised manuscript. All the dots are the same color.

Figure 4, legend: Why is it a five-model mean for the snow concentration? Which models were

not taken into account for the average and why?

Response: GFDL-ESM2M, HadGEM2-ES, and IPSL-CM5A-LR do not provide the surface snow area fraction (snc). We added the sentence to notify this in a Figure 4's caption as follows: (P28-29 L492-495) "All of the values are area-weighted averages of eight ESMs, except for the snow concentration, which is an average of five ESMs, because GFDL-ESM2M, HadGEM2-ES, and IPSL-CM5A-LR do not provide the surface snow area fraction data."

L98-99: This paragraph ends rather abruptly. It would be nice to either provide an explanation of this remote effect (as done for land warming just above) or simply indicate that this is discussed later in the manuscript.

Response: We have added the sentence as follows:

(P6 L110-111) "The mechanisms of this remotely induced Arctic warming will be discussed in the next section."

L106-107: For clarity, consider rephrasing with "... during boreal winter and its magnitude is even comparable ..."

Response: We have rephrased that sentence as follows:

(P6-7 L118-121) "The Arctic warming resulting from the physiological effects is most distinctive during the boreal winter (December–January–February; DJF). In addition, the magnitude of this Arctic warming in DJF is comparable with that of continental warming during JJA."

L110: "A previous study", add "A"

Response: corrected.

L115: "the Pan–Arctic region". It is a bit unclear how this term is used throughout the paper. It seems from the legend in Figure 4 that Pan-Arctic meant $>70^{\circ}\text{N}$, for some people it means $>66^{\circ}\text{N}$ or even lower, maybe this could be better defined in the introduction.

Response: To clarify we tried to use "Arctic region" rather than "pan-Arctic region" and defined the Arctic region in the introduction as follows:

(P3 L46-49) "This physiological effect has a potential to remotely alter the entire climate system through the redistribution of the surface energy and disturbance of hydrological cycle, but still, the remote impacts of physiological effect on the climate system are unclear especially in the Arctic region (north of 70°N)."

L123-125: "different physical processes". This could be more specific. The current explanation of the processes leading to a remote and lagged effect is not very convincing here and could be elaborated. This seems like a missing link in the current manuscript.

Response: We have elaborated the mechanisms for the remote and lagged physiological effects on Arctic warming as follows:

(P7-8 L122-154) "It is important to understand how this continental warming resulting from the physiological effects can remotely cause the distinctive delayed warming in the Arctic Ocean. A previous study demonstrated that mid and high latitude forcing can remotely contribute to the Arctic warming through various physical processes⁴¹. In particular, the increase in atmospheric northward energy transport (NHT_{ATM}) due to continental warming can be responsible for the delayed Arctic warming. It is evident that the northward energy transport significantly increases during the warm season from April to July (see the Methods section and Supplementary Fig. 5a). Furthermore, this NHT_{ATM} continuously increased during the whole period of simulations due to the intensified CO_2 physiological forcing with increasing CO_2

levels (Supplementary Fig. 5b). These results imply that the NHT_{ATM} plays a role in connecting the extratropical continent warming to the Arctic warming under an influence of the physiological effect, which shows the similarity with previous studies, suggesting that the high-latitude greening and mid-latitude afforestation can enhance the Arctic amplification through an increase in poleward energy transport^{40,42–44}. The increase in NHT_{ATM} causes sea ice melting and the resulting newly open waters in the Arctic allow it to absorb more sunlight during the warm season (Fig. 4 and Supplementary Table 6). Most of this energy is released to the atmosphere through the longwave radiative flux, and sensible and latent heat flux in the Arctic Ocean during autumn and winter, thereby inducing the Arctic warming (Supplementary Table 6). These mechanisms, the seasonal storage and release of the absorbed shortwave radiation coupling with an Arctic sea ice change, have already proposed in previous studies to explain the Arctic amplification^{19–21}. However, our results newly suggest that the plant physiological forcing as well as radiative forcing can cause the Arctic amplification under elevated CO₂ levels.

A previous study has shown that the mid-troposphere in the Arctic sensitively responds to the energy advection across the Arctic boundary²⁶. The bottom-heavy warming profile has been attributed to increased upward turbulent heat fluxes by the loss of sea ice in previous studies^{17,19}. The vertical structure of atmospheric warming shows that the mid-tropospheric warming first occurred with a large vertical extent in the Arctic region during summer, and then this warming was propagated to the lowermost region of the atmosphere with time (Supplementary Fig. 6). These results support our hypothesis that the remote and lagged effects of plant physiological acclimation can intensify Arctic warming through an enhancement of NHT_{ATM} and the resulting Arctic sea ice change.”

L125: It is maybe worth mentioning that the remote effects are very strong only in some of the models (e.g. Suppl. Figure 8, bcc-csm-1-1, canESM2, and HadGEM2-ES). Several models show much less propagation to the arctic ocean and little winter effects (e.g. CESM1-BGC, IPSL-CM5A-LR).

Response: We have mentioned the inter-model diversity in magnitude of the Arctic warming resulting from the remote and lagged effects of physiological forcing as follows:

(P8 L154-157) “However, there is a large inter-model diversity in the magnitude of Arctic warming, which seems to be closely related to the strength of local feedback related to Arctic sea ice, but further research is needed to confirm this (Supplementary Figs. 8,12b).”

L133: consider replacing “to the climate system” with “to arctic warming”

Response: We have replaced the word “the climate system” to “Arctic warming” as follows:

(P9 L165-166) “These results emphasize that the contribution of plant physiological effects to the Arctic warming is quite significant.”

L128-133: This paragraph might potentially be moved to the previous section.

Response: We have moved this paragraph to the previous section as follows:

(P8-9 L158-166) “In summary, the surface warming resulting from physiological effect enhances an atmospheric energy convergence into the Arctic basin and this increases the net SW absorption during the warm season in the Arctic Ocean by melting sea ice. Subsequent energy release to the atmosphere increases the ice-free waters in the Arctic, thereby intensifying the Arctic amplification during the cold season. As a result, the CO₂ physiological forcing accounts for 27.7% of the continental warming in summer and 9.7% of the annual surface warming in Arctic region resulting from CO₂ radiative forcing (Fig. 3). These results emphasize that the contribution of the plant physiological effects to the Arctic warming is quite significant.”

L153-155: For those models with interactive vegetation, replacing bare soils with vegetation or tundra with boreal forests might also change the albedo of the surface, independently of changes in temperature that would melt the snow.

Response: We have mentioned this mechanism in the revised manuscript as follows:

(P10 L189-192) “Furthermore, an increase of LAI from the fertilization effects and the land cover change in models with interactive vegetation might partially contribute to the surface warming by altering the surface albedo independently of a change in temperature and would melt the snow as noticed previously^{45,46} (Supplementary Figs. 9,14).”

L156-157: This seems like a plausible hypothesis but the presented data cannot really be used to demonstrate it. Maybe this statement should be moderated.

Response: We have edited these sentences as follows:

(P10 L192-195) “Consequently, this snow–albedo feedback may help enhance and maintain the land surface warming throughout the year, especially in high latitudes where the surface albedo is relatively high due to the high snow cover (Fig. 4e).”

L159: “was mostly consumed in accelerating the melting of sea ice”. I find this statement hypothetical and not supported by data. It could be better demonstrated for instance by computing the amount of energy necessary for the additional melting and comparing it with the additional energy actually advected (that might be even done for each model and used to check if there is a consistency, similar to what is attempted in suppl. Figure 9).

I find the absolute reduction in latent heat flux rather intriguing actually, why would it decrease if there is less sea ice, more open water and warmer air? The decrease in latent heat might actually be due to land ET reduction from parcels of land areas $>70^{\circ}\text{N}$, this would be easy to check.

Response: It is usually known that an additional heat to the ice-covered surface is used for melting sea-ice or is stored in newly open waters in the Arctic Ocean, rather than increasing temperature during the warm season when the ice is melting (Dai et al. 2019). In order to avoid the confusion, we just removed the sentence.

L180-181: Honestly, that relationship is not clear and the regressions are not significant. This statement should be modified.

Response: We deleted the part, which describe inter-model relationship.

L183-186: Again that correlation is not significant ($p>5\%$) according to supplementary figure 9 and mostly relies on HadGEM2-ES having such a strong response.

Response: We deleted the part, which describe inter-model relationship.

L195 “a previous study” add “a”

Response: corrected.

L202. I could not find any indication of this in the mentioned reference. Please indicate precisely which models do not include physiological effects.

Response: We have revised the sentence as follows:

(P11 L229-231) “Currently, several models in the CMIP5 ensemble do not include the dependence of physiological effects on CO_2 concentration⁵¹: CSIRO-Mk3.6.0, INM-CM4, and MRI-CGCM3. “

References

Bathiany, S., Claussen, M., & Brovkin, V. CO₂-induced Sahel greening in three CMIP5 Earth system models. *J. Clim.* **27**, 7163–7184 (2014).

Dai, A., Luo, D., Song, M., & Liu, J. Arctic amplification is caused by sea-ice loss under increasing CO₂. *Nat. commun.* **10**, 121 (2019).

Skinner, C. B., Poulsen, C. J., Chadwick, R., Diffenbaugh, N. S. & Fiorella, R. P. The role of plant CO₂ physiological forcing in shaping future daily-scale precipitation. *J. Clim.* **30**, 2319–2340 (2017).

Reviewers' comments:

Reviewer #1 (Remarks to the Author):

Second review of "The intensification of Arctic warming as a result of CO2 physiological forcing"

The authors have done a thorough job addressing each of my original comments. In particular, they have incorporated new analyses on the relative roles of CO2 fertilization and CO2 physiological forcing on the surface energy balance. They have also corrected their calculation of meridional energy transport. The authors have presented a convincing argument detailing the pathway through which vegetation changes can influence Arctic temperatures. At this stage, I recommend the manuscript for publication in Nature Communications.

Reviewer #2 (Remarks to the Author):

Review for NCOMMS-19-31397A submission from Park et al. entitled "The intensification of Arctic warming as a result of CO2 physiological forcing."

Park and colleagues provide a revision to their work that claims that CO2 reduces boreal ET, leading to a partitioning of energy towards sensible heat, thereby amplifying Arctic warming over the land, and in particular, leads to remote warming of surface air over the ocean via surface and atmospheric albedo feedbacks.

While the authors have provided responses to my comments that I think seem to strengthen their claims somewhat, it is not clear to me how their responses have been explicitly incorporated into their manuscript. In fact, of the 10 additional figures the authors provided to my comments in the response file, I have little sense of how many of them actually made it into the manuscript as figures or as a discussion point. Certainly putting all these robustness checks into the supplemental is not possible, which is fine, but despite that, the authors do not point to where in the manuscript that substantive changes to the text that are actually responsive to the review are made. Only in one response to my comments do they point to a change in the text. I find this odd given the work they have done and it's hard not to see it as a bit dismissive.

I discuss the authors' response to each of my previous comments below:

1. Generalizability: I believe the authors make a convincing case of the robustness across the models on the direction, if not the magnitude, of the warming effect. A simple line in the manuscript attesting to the robustness even if outliers are left out would be sufficient.
2. Context relative to Rad-only and Total effects. The authors did all this work to illustrate the magnitude of the nonlinear interaction between radiation and physiology (equivalent to 24% or 30% of their annual-scale temperature change, which itself is quite modest) and yet this is only referenced as a brief line at 91 (pg. 5), suggesting the results are consistent. Sure, the results are consistent, but the actual effect size of the physiological forcing is 30% lower for the Arctic. That strikes me as something that is worth a more substantive discussion.
3. The ET drop suggests that fertilization effects are smaller than stomatal ones on total ET. The authors appear to have missed my point, which I'll take as my failure to communicate it. The combined effects of radiation and CO2 lead to net increases in ecosystem water consumption, not the relative effects of fertilization and conductance. Yes, conductance in this ensemble seems to win out, but we're not going to be living in a world where vegetation sees CO2, but the planet's radiation budget does not. See the recent Mankin et al. Nature Geoscience 2019. So while in these C4MIP-style experiments it is clear that ET declines due to conductance changes alone (Fig. 1a), it is also clear that

the net effects in both this and the wider CMIP5 ensemble, that ET increases (Fig. S1a). All of the literature the authors cite are about the effect of conductance changes relative to fertilization absent radiation changes, which is not how the real-world works. Ecosystem water consumption increases in the projections irrespective of conductance changes. So the notion that at page 5, line 84, that the fertilization effect is weaker than the stomatal effect is a bit of a red herring, as LAIs are higher, the growing season is longer, and plants are still evapotranspiring more. It would be helpful if the authors crafted their language a bit more carefully that they're really only talking about these effects in this idealized world, not the real (fully-coupled) one.

Reviewer #3 (Remarks to the Author):

My evaluation is that the authors have answered the main points which were raised during the review process. I thank the authors for their efforts in providing all the necessary supplementary material. This study may be suited for publication in Nature Communications. However, there are still some clumsy formulations or structures that can make the paper hard to read. Some of the content also seems repetitive, e.g. L168-208 mostly restate or expand on aspects that are already briefly covered previously in the text. Some of this content could be re-organized and thus the flow of the paper would be more focused. The last paragraph of the conclusion also seems to read like a « train of thought ». It could be polished and made more compelling.

L15 « The stomatal closure » consider replacing with « Stomatal closure ».

L19 consider removing «the presence of ».

L23 replace « causes » with « contributes to ». It is by far not the only cause of sea ice loss.

L24 consider replacing « leading to » with « enhancing ». That would be more accurate.

L26 That sentence structure is odd. Maybe «, and consequently the contribution of physiological effects to Arctic warming represents about 10% of all radiative forcing effects. »

Consider improving the transition. « At the same time, the fertilization... »

L89 « obviously does not »

L98 « a strong seasonality »

L103 : You should reference the plots added in response to the reviewer's comments here (figure A3 of rebuttal letter).

L110 replace « will be » with « are »

L120 « comparable ». But does this also represent a comparable total amount of energy ?

L132 « continental »

L136 suggest to replace « causes » with « is associated with ». The experiment does not explicitly demonstrate the causality. I agree it's a reasonable hypothesis, but there might be other hypotheses to explain that.

L142 « already been proposed »

L144 replace « can cause » with « can contribute to » to avoid any misunderstanding. There are many other drivers causing Arctic amplification under elevated CO2.

L168. The header of this section does not seem very appropriate. This section first focuses on inter-model differences, then discusses snow albedo feedbacks, then comes back to cloud feedbacks again. The flow and logic should be improved.

L173-179 Isn't that already discussed at L103-107 ? It seems repetitive.

L229-231. With CMIP6 around the corner, I would remove this sentence. Also this does not really belong to the broader perspective a conclusion should bring.

Other sentences needing improvement :

- L90-93
- L215-218

Response to Reviewers' comments:

Reviewer #1 (Remarks to the Author):

Second review of "The intensification of Arctic warming as a result of CO₂ physiological forcing"

The authors have done a thorough job addressing each of my original comments. In particular, they have incorporated new analyses on the relative roles of CO₂ fertilization and CO₂ physiological forcing on the surface energy balance. They have also corrected their calculation of meridional energy transport. The authors have presented a convincing argument detailing the pathway through which vegetation changes can influence Arctic temperatures. At this stage, I recommend the manuscript for publication in Nature Communications.

Response: We appreciate the Reviewer #1's encouraging comment.

Reviewer #2 (Remarks to the Author):

Review for NCOMMS-19-31397A submission from Park et al. entitled “The intensification of Arctic warming as a result of CO2 physiological forcing.”

Park and colleagues provide a revision to their work that claims that CO2 reduces boreal ET, leading to a partitioning of energy towards sensible heat, thereby amplifying Arctic warming over the land, and in particular, leads to remote warming of surface air over the ocean via surface and atmospheric albedo feedbacks.

While the authors have provided responses to my comments that I think seem to strengthen their claims somewhat, it is not clear to me how their responses have been explicitly incorporated into their manuscript. In fact, of the 10 additional figures the authors provided to my comments in the response file, I have little sense of how many of them actually made it into the manuscript as figures or as a discussion point. Certainly putting all these robustness checks into the supplemental is not possible, which is fine, but despite that, the authors do not point to where in the manuscript that substantive changes to the text that are actually responsive to the review are made. Only in one response to my comments do they point to a change in the text. I find this odd given the work they have done and it’s hard not to see it as a bit dismissive.

Response: We thank the reviewer #2 for helpful and detailed comments. Based on the reviewers’ comments, we tried to reflect our further analyses more to the revised manuscript. Our responses to the specific comments are as follows:

I discuss the authors’ response to each of my previous comments below:

1. Generalizability: I believe the authors make a convincing case of the robustness across the models on the direction, if not the magnitude, of the warming effect. A simple line in the manuscript attesting to the robustness even if outliers are left out would be sufficient.

Response: We have briefly mentioned this in the revised manuscript and also added the supplementary figures in the revised Supplementary Information in detail as follows: (P11 L221-232) “The magnitudes and spatial patterns of change in ET and temperature are diverse and HadGEM2-ES seems to greatly contribute to the multi-model ensemble mean temperature change (Supplementary Figs. 6,7). Nevertheless, most models consistently simulate the reduction in ET, the resulting surface warming over the continents and enhanced Arctic warming as a result of physiological effect (Supplementary Fig. 10), which suggests that the results are not sensitive to a subsampling of the models. In addition, multi-model ensemble results excluding HadGEM2-ES are not much different with those including HadGEM2-ES and still statistically significant though the magnitude is a bit altered (Supplementary Figs. 15,16). These again attest to the robustness of our findings and also suggest that ensemble mean is not controlled by an outlier.”

2. Context relative to Rad-only and Total effects. The authors did all this work to illustrate the magnitude of the nonlinear interaction between radiation and physiology (equivalent to 24% or 30% of their annual-scale temperature change, which itself is quite modest) and yet this is only referenced as a brief line at 91 (pg. 5), suggesting the results are consistent. Sure, the results are consistent, but the actual effect size of the physiological forcing is 30% lower for the Arctic. That strikes me as something that is worth a more substantive discussion.

Response: We have added the discussion of the quantitative evaluation of a synergy effect and the supplementary table in the revised manuscript as follows:

(P5-6 L94-103) “Another interesting point is that a synergy effect, a nonlinear interaction of physiological forcing with the radiative forcing^{15,37}, additionally contributes to the surface warming (see the Methods section and Supplementary Table 7). The magnitude of synergy effect in Arctic region is equivalent to ~24% of annual mean temperature change resulting from physiological forcing. These results imply that the global warming signal by the radiation forcing plays a role in amplifying the physiological effect through their interactions. Meanwhile, the physiological forcing excluding a synergy effect still induces the statistically significant Arctic warming, which confirms the consistency and robustness of our findings (Supplementary Fig. 2).”

3. The ET drop suggests that fertilization effects are smaller than stomatal ones on total ET. The authors appear to have missed my point, which I'll take as my failure to communicate it. The combined effects of radiation and CO₂ lead to net increases in ecosystem water consumption, not the relative effects of fertilization and conductance. Yes, conductance in this ensemble seems to win out, but we're not going to be living in a world where vegetation sees CO₂, but the planet's radiation budget does not. See the recent Mankin et al. Nature Geoscience 2019. So while in these C4MIP-style experiments it is clear that ET declines due to conductance changes alone (Fig. 1a), it is also clear that the net effects in both this and the wider CMIP5 ensemble, that ET increases (Fig. S1a). All of the literature the authors cite are about the effect of conductance changes relative to fertilization absent radiation changes, which is not how the real-world works. Ecosystem water consumption increases in the projections irrespective of conductance changes. So the notion that at page 5, line 84, that the fertilization effect is weaker than the stomatal effect is a bit of a red herring, as LAIs are higher, the growing season is longer, and plants are still evapotranspiring more. It would be helpful if the authors crafted their language a bit more carefully that they're really only talking about these effects in this idealized world, not the real (fully-coupled) one.

Response: We agree with the reviewer that we misinterpreted the reviewer's comment. We totally agree that the combined effects of radiation and physiology lead to net increase in ET. As the reviewer suggested, we tried to argue our point more carefully as follow:

(P5 L81-87) “In this idealized experiment for evaluating the CO₂ physiological forcing, the fertilization effect plays a role in increasing ET due to the resulting increased LAI, but the effect of stomatal closure works in the opposite way at the same time^{5,10,37}. Therefore, this overall drop in ET suggests that the stomatal closure have a greater influence in controlling the total ET than the CO₂ fertilization, when only the physiological effects is considered under elevated CO₂ levels, in consistency with the argument in previous studies^{12,31,34,37}.”

Reviewer #3 (Remarks to the Author):

My evaluation is that the authors have answered the main points which were raised during the review process. I thank the authors for their efforts in providing all the necessary supplementary material. This study may be suited for publication in Nature Communications.

However, there are still some clumsy formulations or structures that can make the paper hard to read. Some of the content also seems repetitive, e.g. L168-208 mostly restate or expand on aspects that are already briefly covered previously in the text. Some of this content could be re-organized and thus the flow of the paper would be more focused.

Response: We thank the reviewer #3's encouraging comments and careful reading. The reviewer's comments were very helpful for significantly improving the manuscript. The reviewer's comments were fully incorporated in the revised manuscript. Our responses to the specific comments are as follows:

The last paragraph of the conclusion also seems to read like a « train of thought ». It could be polished and made more compelling.

Response: We have carefully rewritten this paragraph as follows:

(P11-12 L232-249) “This study has shown that the physiological effects amplify Arctic warming by 9.7% compared with that from the radiative forcing. This surface warming in the Arctic region resulting from the physiological response might have the potential ramifications of future changes in the carbon and hydrological cycles by intensifying the interaction between the Arctic climate and Arctic biological system^{47,48}. Considering the physiological effects of CO₂ might be helpful for understanding the inter-model diversity in future climate change. A previous study has reported that the stomatal conductance schemes in the current ESMs do not consider various plant water use strategy⁴⁹, which can lead to the underestimation of the surface warming across Northern Eurasia⁵⁰. This result raises a possibility that Arctic warming may be greater than that in the current projections. Furthermore, there are still the limitations of land surface models in simulating LAI and the albedo dynamics and the stomatal conductance schemes in ESMs are rather static and semi-empirical (see the Supplementary Notes). These factors make it hard to simulate the realistic plant behavior to elevated CO₂ levels and also increase the uncertainty in the quantification of climate change caused by the physiological forcing. These point to the need for improvement of land models' schemes based on a fundamental understanding of the involved processes.”

L15 « The stomatal closure » consider replacing with « Stomatal closure ».

Response: Replaced.

L19 consider removing «the presence of ».

Response: Deleted

L23 replace « causes » with « contributes to ». It is by far not the only cause of sea ice loss.

Response: Replaced.

L24 consider replacing « leading to » with « enhancing ». That would be more accurate.

Response: Replaced

L26 That sentence structure is odd. Maybe «, and consequently the contribution of physiological effects to Arctic warming represents about 10% of all radiative forcing effects.»

Response: We have revised the sentence as follows:

(P2 L24-27) “The surface warming in the Arctic is further amplified by local feedbacks, and consequently the contribution of physiological effects to Arctic warming represents about 10% of radiative forcing effects.”

Consider improving the transition. « At the same time, the fertilization... »

Response: We have revised the sentence as follows:

(P5 L81-84) “In this idealized experiment for evaluating the CO₂ physiological forcing, the fertilization effect plays a role in increasing ET due to the resulting increased LAI, but the effect of stomatal closure works in the opposite way at the same time^{5,10,37}.”

L89 « obviously does not »

Response: Corrected.

L98 « a strong seasonality »

Response: Corrected.

L103 : You should reference the plots added in response to the reviewer’s comments here (figure A3 of rebuttal letter).

Response: To re-organize the last section in results, we have revised this sentence and also properly referred this supplementary figure (Supplementary Fig. 12) as follows:

(P9 L172-183) “Besides the direct heating from the enhanced sensible heat flux, an increase in net shortwave absorption (4.58 W m^{-2} in JJA) additionally heats the air above the surface in JJA (Supplementary Table 5). In this experimental design, the net SW absorption can be largely affected by these two factors: An increase in LAI resulting from CO₂ fertilization effect can alter the surface albedo and increase the net SW absorption, thereby contributing to the temperature rise. The decrease in cloud fractions caused by physiological acclimation-driven reduction of relative humidity^{35,43,44} can also be a cause of surface warming because it enhances downward SW radiative flux⁴². From their relative contributions, we found that vegetation-cloud feedback has a dominant role in the increased net SW absorption during summer (Supplementary Fig. 12), thereby contributing the continental warming (40°–70°N) particularly in summer (Fig. 4, Supplementary Fig. 13).”

L110 replace « will be » with « are »

Response: Replaced.

L120 « comparable ». But does this also represent a comparable total amount of energy ?

Response: In mid-to-high latitudes (40°–70°N), the surface temperature is ~1.05K and total surface energy flux, defined as the sum of net SW flux, net LW flux, latent heat flux, and sensible heat flux, is ~3.4 W m^{-2} during boreal summer. In Arctic region (70°–90°N), the surface temperature is ~0.99K and total surface energy flux is ~2.54 W m^{-2} during boreal winter.

L132 « continental »

Response: Corrected.

L136 suggest to replace « causes » with « is associated with ». The experiment does not explicitly demonstrate the causality. I agree it's a reasonable hypothesis, but there might be other hypotheses to explain that.

Response: Replaced.

L142 « already been proposed »

Response: Corrected.

L144 replace « can cause » with « can contribute to » to avoid any misunderstanding. There are many other drivers causing Arctic amplification under elevated CO₂.

Response: Corrected.

L168. The header of this section does not seem very appropriate. This section first focuses on inter-model differences, then discusses snow albedo feedbacks, then comes back to cloud feedbacks again. The flow and logic should be improved.

Response: We have re-organized this section. We first focus on the vegetation-cloud feedback, then discuss the cloud feedback in Arctic region and then the snow-albedo feedback. In this section, we only focus on the local feedbacks and moved the discussion about the inter-model diversity to the discussion section as follows:

(P9-11 L172-215) “Besides the direct heating from the enhanced sensible heat flux, an increase in net shortwave absorption (4.58 W m^{-2} in JJA) additionally heats the air above the surface in JJA (Supplementary Table 5). In this experimental design, the net SW absorption can be largely affected by these two factors: An increase in LAI resulting from CO₂ fertilization effect can alter the surface albedo and increase the net SW absorption, thereby contributing to the temperature rise. The decrease in cloud fractions caused by physiological acclimation-driven reduction of relative humidity^{35,43,44} can also be a cause of surface warming because it enhances downward SW radiative flux⁴². From their relative contributions, we found that vegetation-cloud feedback has a dominant role in the increased net SW absorption during summer (Supplementary Fig. 12), thereby contributing the continental warming (40° – 70° N) particularly in summer (Fig. 4, Supplementary Fig. 13). Furthermore, the relative magnitude of the vegetation-cloud feedback in ESMs seems to explain the inter-model diversity of the land surface warming (40° – 70° N) in JJA ($r=-0.79$, $P=0.02$) (Supplementary Fig. 11a). Specifically, two models, HadGEM2-ES and MPI-ESM-LR, having the largest increase of downward SW radiation, show the greatest warming in JJA due to this greatest cloud effect despite the moderate reduction of ET (Supplementary Table 8 and Figs. 6,7,9).

In contrast to the change of cloud cover over the continents (40° – 70° N), the cloud formation is enhanced in the Arctic region especially during winter (Fig. 4 and Supplementary Table 6). This increased cloud fraction additionally intensifies the surface warming by decreasing the outgoing longwave radiation especially in non-summer season^{45,46} (Supplementary Table 6). Although it is difficult to prove the causality in this experiment, it is conceived that this increase in cloud formation contributes to the Arctic sea ice loss, which in turn causes the increase in water vapor from the newly opened Arctic waters, as proposed previously^{21,46}. In summary, the cloud feedback in the Arctic can enhance the surface warming by increasing a downward LW radiation, and in turn, the enhanced surface warming can accelerate the sea-ice loss, thereby causing positive feedback during the cold season.

Another local feedback might be triggered by physiological forcing over the continents (40° – 70° N). As shown in Fig. 4, a snow concentration and a surface albedo in high

latitudes significantly decline in response to the CO₂ physiological forcing. The warming resulting from the physiological effects presumably melts snow and the resultant less-snow covered surface absorbs more solar radiation (Supplementary Table 5). Furthermore, an increase of LAI from the fertilization effects and the land cover change in models with interactive vegetation might partially contribute to the surface warming by altering the surface albedo independently of a change in temperature and would melt the snow as noticed previously^{47,48} (Supplementary Figs. 8,14). Consequently, this snow–albedo feedback may help enhance and maintain the land surface warming throughout the year, especially in high latitudes where the surface albedo is relatively high due to the high snow cover (Fig. 4e). On the whole, our results suggest that the local feedbacks triggered by physiological effects might additionally contribute to the amplified and maintained surface warming in both continents and Arctic Ocean.”

L173-179 Isn't that already discussed at L103-107 ? It seems repetitive.

Response: We have removed the L103-107 and have discussed this vegetation-cloud feedback in the last section of the results.

L229-231. With CMIP6 around the corner, I would remove this sentence. Also this does not really belong to the broader perspective a conclusion should bring.

Response: We have removed that sentence.

Other sentences needing improvement :

- L90-93
- L215-218

Response: We have revised these sentences as follows:

(P6 L100-103) “Meanwhile, the physiological forcing excluding a synergy effect still induces the statistically significant Arctic warming, which confirms the consistency and robustness of our findings (Supplementary Fig. 2).”

(P11 L223-228) “Nevertheless, most models consistently simulate the reduction in ET, the resulting surface warming over the continents and enhanced Arctic warming as a result of physiological effect (Supplementary Fig. 10), which suggests that the results are not sensitive to a subsampling of the models.”

REVIEWERS' COMMENTS:

Reviewer #2 (Remarks to the Author):

Review for NCOMMS-19-31397B submission from Park et al. entitled "The intensification of Arctic warming as a result of CO2 physiological forcing."

The authors have addressed all of my comments, and now make clear where in the manuscript they have been responsive to the reviews.